# Facet-engineered TiO$_2$ drives photocatalytic activity and stability of supported noble metal clusters during H$_2$ evolution

Yufen Chen [1,2], Lluís Soler [1,2] ✉, Claudio Cazorla [3], Jana Oliveras[4], Neus G. Bastús[4], Víctor F. Puntes[4,5,6] & Jordi Llorca [1,2] ✉

Metal clusters supported on TiO$_2$ are widely used in many photocatalytic applications, including pollution control and production of solar fuels. Besides high photoactivity, stability during the photoreaction is another essential quality of high-performance photocatalysts, however systematic studies on this attribute are absent for metal clusters supported on TiO$_2$. Here we have studied, both experimentally and with first-principles simulation methods, the stability of Pt, Pd and Au clusters prepared by ball milling on nanoshaped anatase nanoparticles preferentially exposing {001} (plates) and {101} (bipyramids) facets during the photogeneration of hydrogen. It is found that Pt/TiO$_2$ exhibits superior stability than Pd/TiO$_2$ and Au/TiO$_2$, and that {001} facet-based photocatalysts always are more stable than their {101} analogous regardless of the considered metal species. The loss of stability associated with cluster sintering, which is facilitated by the transfer of photoexcited carriers from the metal species to the neighbouring Ti and O atoms, most significantly and detrimentally affects the H$_2$-evolution photoactivity.

Anatase titania (TiO$_2$) has been widely employed for photocatalytic production of hydrogen (H$_2$) due to its suitable conduction and valence band edges and superior photostability. Huge efforts, though, have been devoted to optimizing the photocatalytic performance of TiO$_2$ by modifying its physical properties and/or blending it with metal species[1–4]. It is generally accepted that the {101} TiO$_2$ facet is thermodynamically more stable than the {001} facet because of its lower average surface energy, 0.44 J/m$^2$ and 0.90 J/m$^2$, respectively[5,6]. The photocatalytic activity of TiO$_2$ is highly facet and shape dependent and extensive studies have investigated the influence of specific surface coordination environments on it[7,8].

Determining which anatase TiO$_2$ facet is photocatalytically the most active in fact remains contentious. For instance, the {001} facet

has been revealed to be more efficient than the {101} facet[9,10] because it contains 100% 5-fold-coordinated titanium atoms (5c-Ti), which renders more active sites and stronger interactions with the adsorbates during the photocatalytic reactions[11], whereas the {101} surface facet possesses 50% 5c-Ti and 50% 6c-Ti[5,12]. Conversely, the {101} facet showed higher photocatalytic H$_2$ evolution than the {001} facet in an aqueous methanol solution[10,13]; in this latter case, however, platinum (1 wt%) was deposited onto the TiO$_2$ samples as a co-catalyst. The co-catalyst plays an important role in the photocatalytic phenomenon, not only by improving the overall photocatalytic performance, but also by showing different trends in the H$_2$ evolution rates depending on the facets exposed by TiO$_2$[14]. It has been demonstrated that the presence of an optimum ratio of {101}/{001} facets in TiO$_2$ anatase is capable of

[1]Department of Chemical Engineering, Universitat Politècnica de Catalunya, Eduard Maristany 16, EEBE, Barcelona 08019, Spain. [2]Institute of Energy Technologies and Barcelona Research Center in Multiscale Science and Engineering, Universitat Politècnica de Catalunya, Eduard Maristany 16, EEBE, Barcelona 08019, Spain. [3]Department of Physics, Universitat Politècnica de Catalunya, Campus Nord, B4-B5, Barcelona E-08034, Spain. [4]Institut Català de Nanociència i Nanotecnologia (ICN2), CSIC and The Barcelona Institute of Science and Technology (BIST), Campus UAB, 08193 Barcelona, Spain. [5]Institució Catalana de Recerca I Estudis Avançats (ICREA), 08010 Barcelona, Spain. [6]Vall d'Hebron Research Institute (VHIR), Hospital Universitari Vall d'Hebron, Passeig de la Vall d'Hebron, 129, Barcelona 08035, Spain. ✉e-mail: lluis.soler.turu@upc.edu; jordi.llorca@upc.edu

reducing the recombination of photogenerated carriers due to the formation of a heterojunction between the two different facets, which drives the migration of electrons and holes to the {101} and {001} facets, respectively[15–17]. This optimal {101}/{001} ratio is highly dependent on the studied reaction[6,17–19]. Taken together, these results suggest that the existing ambiguities seem to be caused by many different factors that come into play when evaluating the catalytic activity (e.g., distinct surface terminations, noble metals employed as co-catalysts[14], and even different reactions).

To date, most efforts to improve the photocatalytic activity of TiO₂ have been directed toward synthesizing highly exposed anatase surfaces[9,20] while stabilizing the shape[21,22] of the crystal. It is well known that the photocatalytic activity of noble metal-loaded TiO₂ heavily depends on the interactions between the co-catalysts and TiO₂ support, which are specific to the metal species and anatase surfaces[23–26]. Therefore, to decipher the role of facets in a metal-supported heterogeneous photocatalytic system, both the metals and facets should be carefully considered. Many previous studies have successfully synthesized and evaluated various combinations of noble metals and exposed TiO₂ facets, yet another key factor has been largely neglected: the stability of the metal nanoparticles during the photoreaction[25,27].

Single metal atoms and clusters anchored on exposed anatase facets have been thoroughly analysed due to their abundant active surface area, low-coordination environment and distinctive electronic properties[28,29]. These ultra-small metal entities are prone to aggregate when irradiated with light under reductive conditions due to their high surface free energy, thereby reducing their surface-to-bulk atoms ratio and, in turn, photocatalytic performance[30]. While the agglomeration or sintering of ultra-small particles during high-temperature treatments has been commonly interpreted in terms of Ostwald ripening and particle diffusion[31,32], the underlying causes of cluster coalescence in photoreactions still remain obscure. Despite various attempts to isolate single atoms/clusters during photocatalytic processes, such as zeolite confinement[33], thiolate-protection[34,35], and polymer stabilization[36], there is little holistic understanding of the aggregation phenomena occurring in different metal-supported systems under the same photocatalytic conditions.

Here, we used a mechanochemical strategy to obtain different metal clusters supported on different TiO₂ facets and investigated their long-term stability in photocatalytic hydrogen production. Mechanochemical approaches represent a promising alternative to conventional wet chemistry methods to produce supported single atoms and clusters[37–39]. In particular, Au, Pd and Pt clusters (1 wt%) were dispersed onto anatase plates or bipyramids preferentially exposing {001} or {101} facets, respectively (Supplementary Fig. 1) by a one-step ball milling procedure and their room-temperature photocatalytic performance on the hydrogen evolution reaction in gas phase from a water/ethanol mixture was systematically explored under

dynamic conditions. Our study reveals that, during photoreaction, the noble-metal clusters may either undergo substantial rearrangements and coalescence or remain stable and practically maintain their as-synthesized morphology, depending on the TiO₂ nanoshape. This behaviour is rationalized by state-of-the-art first-principles simulations that effectively account for photoinduced electronic excitations in crystals. Thus, the present work shows that the anatase TiO₂ nanoshape mainly drives the stability and photocatalytic performance of noble metal clusters in H₂ production, a finding that has the potential to be generalised to other types of reactions and probably also to other supports (e.g., rutile TiO₂ and CeO₂).

## Results

TEM images confirmed that anatase TiO₂ plates and bipyramids were succesfully synthesized (Supplementary Fig. 1). Both anatase nanoshapes exhibited {001} and {101} terminations, with plates preferentially exposing {001} facets and bipyramids preferentially exposing {101} facets (see the methods section for details). The size distribution is shown in Supplementary Fig. 2; the average side length and thickness of TiO₂-001 (anatase plates) are 31.1 and 9.8 nm, respectively, and those of TiO₂-101 (anatase bipyramids) are 12.3 and 23.6 nm, respectively. XRD (Supplementary Fig. 3) and Raman spectroscopy (Supplementary Fig. 4) demonstrated that the crystallinity and morphology of TiO₂ were preserved after ball milling and calcination (500 °C, 1 h)[38,40]. After the mechanochemical procedure, the specific surface area (SSA$_{BET}$, Supplementary Table 1) was measured to be, in both cases, ca. 55 m² g⁻¹.

### Photocatalytic H₂ evolution

To evaluate the photocatalytic properties of the different metal clusters-TiO₂ architectures, 1 h, 4 × 1 h light on-off cycle experiments, and 20 h (stability) tests of H₂ production under UV light (365 ± 5 nm) were independently performed (Fig. 1 and Supplementary Fig. 5). The scheme of the continuous flow photoreaction system is shown in Supplementary Fig. 6; products were continuously monitored by micro-gas chromatography. The temperature of the photocatalysts was monitored during the photoreaction and it was always below 54 °C. As shown in Fig. 1a, H₂ was produced immediately after UV radiation was shed on the gaseous reactants, achieving different rates of H₂ evolution depending on the noble metal and anatase nanoshape. The Pt/TiO₂-101 photocatalyst exhibited a remarkable performance with a much higher value of H₂ evolution than that of others. Meanwhile, the Pd/TiO₂-101 photocatalyst showed negligible photoactivity enhancement compared to bare TiO₂, whereas the Pt/TiO₂-001 and Au/TiO₂-101 samples behaved very similarly during the reaction. Overall, the rate of H₂ production for the as-prepared photocatalysts decreased in the order: Pt/TiO₂-101 ≫ Pd/TiO₂-001 > Pt/TiO₂-001 ~ Au/TiO₂-101 ≫ Au/TiO₂-001 > Pd/TiO₂-101 > TiO₂-001 ~ TiO₂-101.

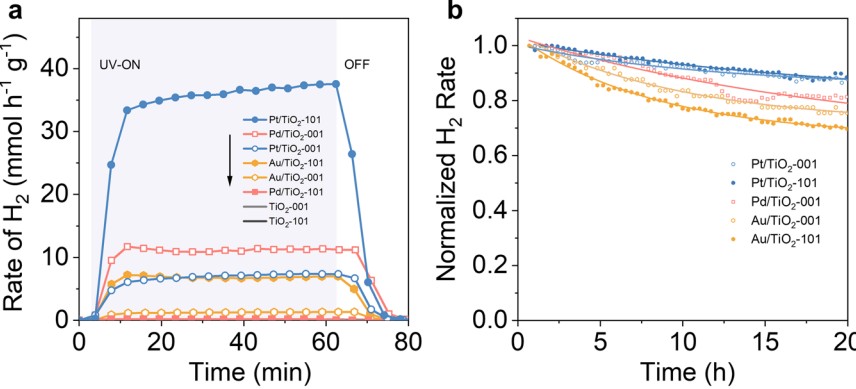

**Fig. 1 | Photocatalytic performance of as-prepared photocatalysts. a** 1 h test of photocatalytic H₂ production. **b** Normalized H₂ evolution rate for 20-h test.

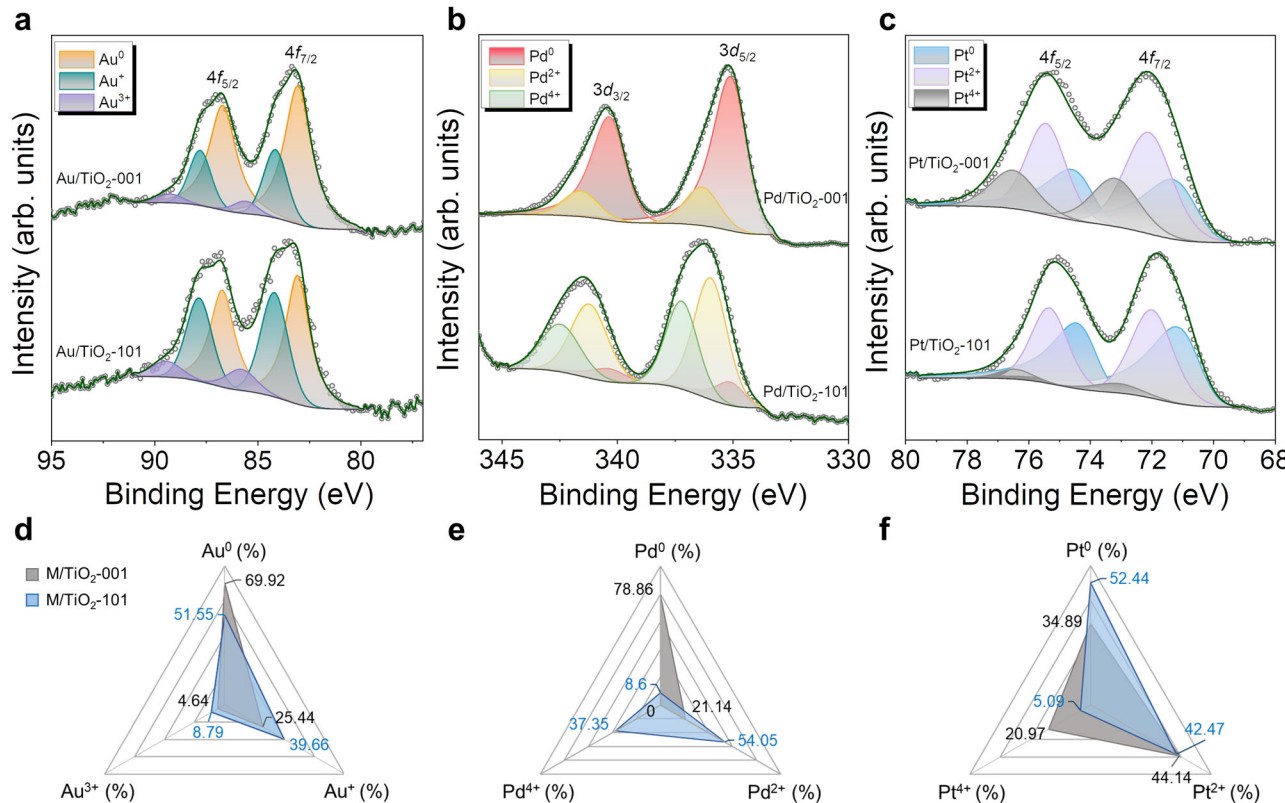

**Fig. 2 | XPS results of Au/TiO$_2$, Pd/TiO$_2$ and Pt/TiO$_2$ photocatalysts. a–c** XP Au $4f$, Pd $3d$ and Pt $4f$ spectra of fresh Au/TiO$_2$-001, Au/TiO$_2$-101, Pd/TiO$_2$-001, Pd/TiO$_2$-101, Pt/TiO$_2$-001 and Pt/TiO$_2$-101 photocatalysts. **d–f** Corresponding ratios of metal oxidation states extracted from XP spectra in (**a–c**).

Interestingly, an increasing trend of H$_2$ evolution was observed for the Pt/TiO$_2$ samples, which can be attributed to a progressive Pt reduction under photoreaction conditions. This phenomenon has been reported by Dessal et al.[41], and later studied by operando X-ray absorption spectroscopy by Piccolo et al.[42]. It is important to recall that the photoactivity trend may be influenced not only by the preferential facet exposed by anatase but also by a more complex situation derived from {001}/{101} interface contributions.

Cycling experiments (Supplementary Fig. 5a) demonstrated good reproducibility of all photocatalysts in short-term photocatalytic processes, but long-term stability experiments (20 h) showed important differences among them (Supplementary Fig. 5b). For comparison purposes, we normalized the H$_2$ evolution rates in Fig. 1b. Our results reveal that Pt clusters loaded on anatase plates and bipyramids exhibited remarkable and identical durability. On the other hand, the Au/TiO$_2$ photocatalysts showed poor stability of H$_2$ production rates for both anatase plates and bipyramids (especially in the bipyramids case). Meanwhile, the stability of the Pd/TiO$_2$-001 sample was in between those of the Pt and Au systems (Pd/TiO$_2$-101 was not considered in the normalized analysis due to its negligible H$_2$ production). Based on these experimental findings, we may conclude that the variability in the photocatalytic performance of functionalized anatase does not solely respond to individual factors like the noble metal species or anatase nanoshape but also the interactions between them. In terms of stability, the measured trend is Pt/TiO$_2$-001 ~ Pt/TiO$_2$-101 > Pd/TiO$_2$-001 > Au/TiO$_2$-001 > Au/TiO$_2$-101, from which it is clear that the metals exhibit different photocatalytic stability, namely, Pt/TiO$_2$ ≫ Pd/TiO$_2$ > Au/TiO$_2$. These results suggest that minor rearrangements of the Pt species take place during the photocatalytic process whereas important migration of Au species occurs, being Pd species an intermediate case. Deactivation caused by metal diffusion into the TiO$_2$ lattice can be safely excluded due to the low temperature of the

experiments (below 54 °C)[43]. Evidence of the unstable Au was also observed in a shell-like gold structure around dihexagonal pyramidal cadmium selenide nanocrystals[44,45]. On the other hand, metals supported on anatase plates are always more stable than those on anatase bipyramids, regardless of the metal considered.

## Characterization of photocatalysts

The oxidation states of Au, Pd and Pt clusters over different anatase nanoshapes were clarified by X-ray photoelectron spectroscopy (XPS). Remarkable differences in the relative distribution of atomic fractions for each supported metal were identified, as shown in Fig. 2. For Au/TiO$_2$ photocatalysts, the fraction of oxidized Au (binding energy of $4f_{7/2}$ for reduced Au: 83.1 eV; Au$^+$: 84.2 eV; Au$^{3+}$: 85.8 eV) was higher in the {101} case than in the {001} surface. It has been reported that oxidized Au species strongly interact with the surface of TiO$_2$, thus accelerating the $e^-$ transfer between the metal and support interface during photocatalysis[46]. These results align with the higher H$_2$ evolution rate obtained for the Au/TiO$_2$-101 compared to Au/TiO$_2$-001. Interestingly, in the Pd/TiO$_2$-001 sample, a large amount of reduced Pd ($3d_{5/2}$ at 335.1 eV, 78.86%) was measured, whereas the dominant Pd species in Pd/TiO$_2$-101 was in the oxidized form (54.05% of Pd$^{2+}$ at 336.3 eV and 37.35% of Pd$^{4+}$ at 337.2 eV). This significant difference in the most abundant Pd oxidation states can be directly ascribed to the distinct photocatalytic performance observed for the {001} and {101} surfaces; the reduced Pd clusters have been proved to be more favourable for photocatalytic H$_2$ production than the high-valent states[38]. In the case of Pt clusters, we observed that the {001} facet induces more Pt$^{4+}$ ($4f_{7/2}$ at 73.2 eV) than the {101} facet, which is less active than the Pt$^0$ (70.8 eV) and Pt$^{2+}$ (72.1 eV) species[47]. Based on the XPS results, we may conclude that different anatase nanoshapes induce different oxidation states in the metal clusters and that this effect depends on the metal species and determines the photocatalytic performance. XPS analyses were

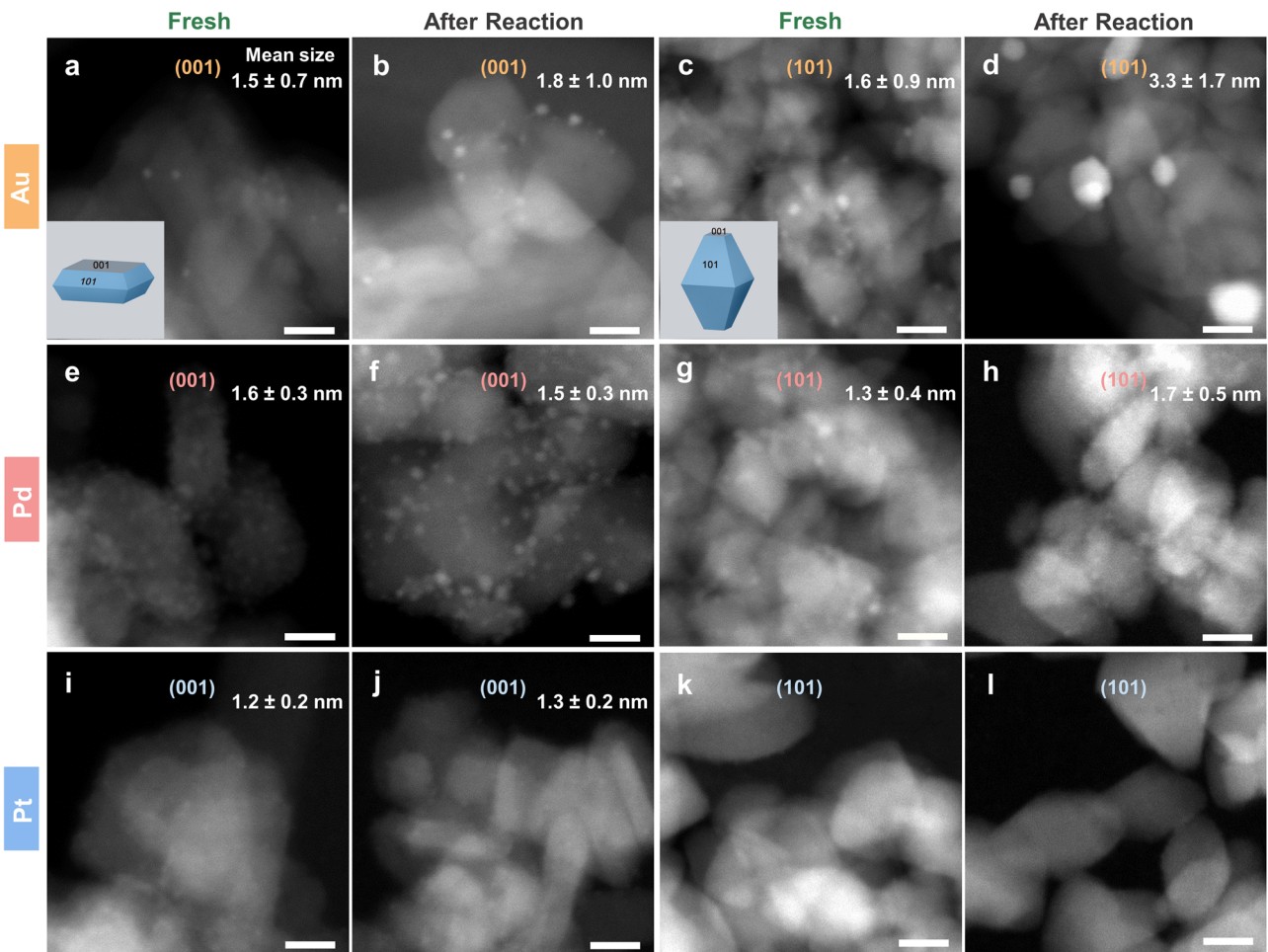

**Fig. 3 | HAADF-STEM images of fresh and after photoreaction samples. a, b** Au/TiO$_2$-001. **c, d** Au/TiO$_2$-101. **e, f** Pd/TiO$_2$-001. **g, h** Pd/TiO$_2$-101. **i, j** Pt/TiO$_2$-001. **k, l** Pt/TiO$_2$-101. Scale bar: 10 nm. The bright dots refer to the corresponding metal clusters in each photocatalyst.

conducted for the post-reacted Au/TiO$_2$-001, Au/TiO$_2$-101, Pd/TiO$_2$-001, Pd/TiO$_2$-101, Pt/TiO$_2$-001 and Pt/ TiO$_2$-101 photocatalysts, as shown in Supplementary Fig. 7. In both Au/TiO$_2$-001 and Au/TiO$_2$-101, Au was reduced after long-term UV irradiation. The metallic Au species (Au$^0$) in Au/TiO$_2$-001 and Au/TiO$_2$-101 increased from 69.9% and 51.5% to 80.9% and 69.4%, respectively. The oxidation states of Pd components in Pd/TiO$_2$-001 exhibited similar proportions before and after photoreaction, whereas Pd/TiO$_2$-101 also underwent reduction, as was similarly observed for Pt/TiO$_2$ photocatalysts. The change in the oxidation states of metal clusters during photocatalysis can be accounted for the reduction of the metal species induced by the photogenerated electrons under the ethanol/water and UV irradiation conditions, as observed in previous studies[2,42].

HAADF-STEM and HRTEM images were acquired to investigate the morphological transformation of the supported metal clusters from freshly synthesized to after 20-h photoreaction (Fig. 3 and Supplementary Fig. 8, and Fig. 4 and Supplementary Fig. 9, respectively). Enlarged HRTEM images of Pt/TiO$_2$ samples are shown in Supplementary Figs. 10 and 11 for a better visualization of the Pt clusters. Representative HAADF-STEM-EDX analysis for Pd/TiO$_2$ (Supplementary Fig. 12) clearly shows the correlation between the metal and the structures identified as metal clusters in the samples. The HAADF-STEM and HRTEM images indicate that there is no preferential deposition of metal clusters on different titania facets, neither in the fresh samples nor in the post-reacted ones. This is attributed to the dry mechanochemical preparation method used, which, in contrast to wet chemistry methods[48–51], does not yield any preferential deposition. The

corresponding cluster size distribution histograms are shown in Supplementary Figs. 13 and 14. The mean size of Au clusters in fresh Au/TiO$_2$-001 and Au/TiO$_2$-101 were about 1.5 nm and 1.6 nm, respectively. They both underwent particle growth after photoreaction, resulting in larger particles of about 1.8 nm and 3.3 nm, respectively. It is evident that the Au species loaded on the {101} surface suffered from more intense agglomeration during the photoreaction among all catalysts, which is perfectly consistent with the worst stability of Au/TiO$_2$-101. The HRTEM images also show that the size of the Pd clusters on the {001} facet increased slightly after photoreaction, in accordance with its moderate stability. By contrast, Pt clusters on both the {001} and {101} surfaces showed similar size distributions before and after the photoreaction, according with their superior stability. Therefore, deactivation can be correlated well with metal sintering, which, in turn, depends on the facet exposed by TiO$_2$.

**First-principles DFT calculations**

To elucidate the mechanisms of metal-cluster stability on TiO$_2$ surfaces, we performed first-principles calculations based on density functional theory (DFT). We estimated the adsorption energy and agglomeration tendency of Au, Pd and Pt atoms and nanoclusters on the {001} and {101} surfaces of anatase (Fig. 5a–c and Supplementary Fig. 15) with the finding that Pt renders the largest adsorption energy and lowest clustering tendency (especially in the {001} facet), followed by Pd and Au atoms. The adsorption energy of Pt and Au nanoclusters was very similar to those obtained for individual atoms, thus hereafter we reasonably extend the conclusions obtained for single atoms to

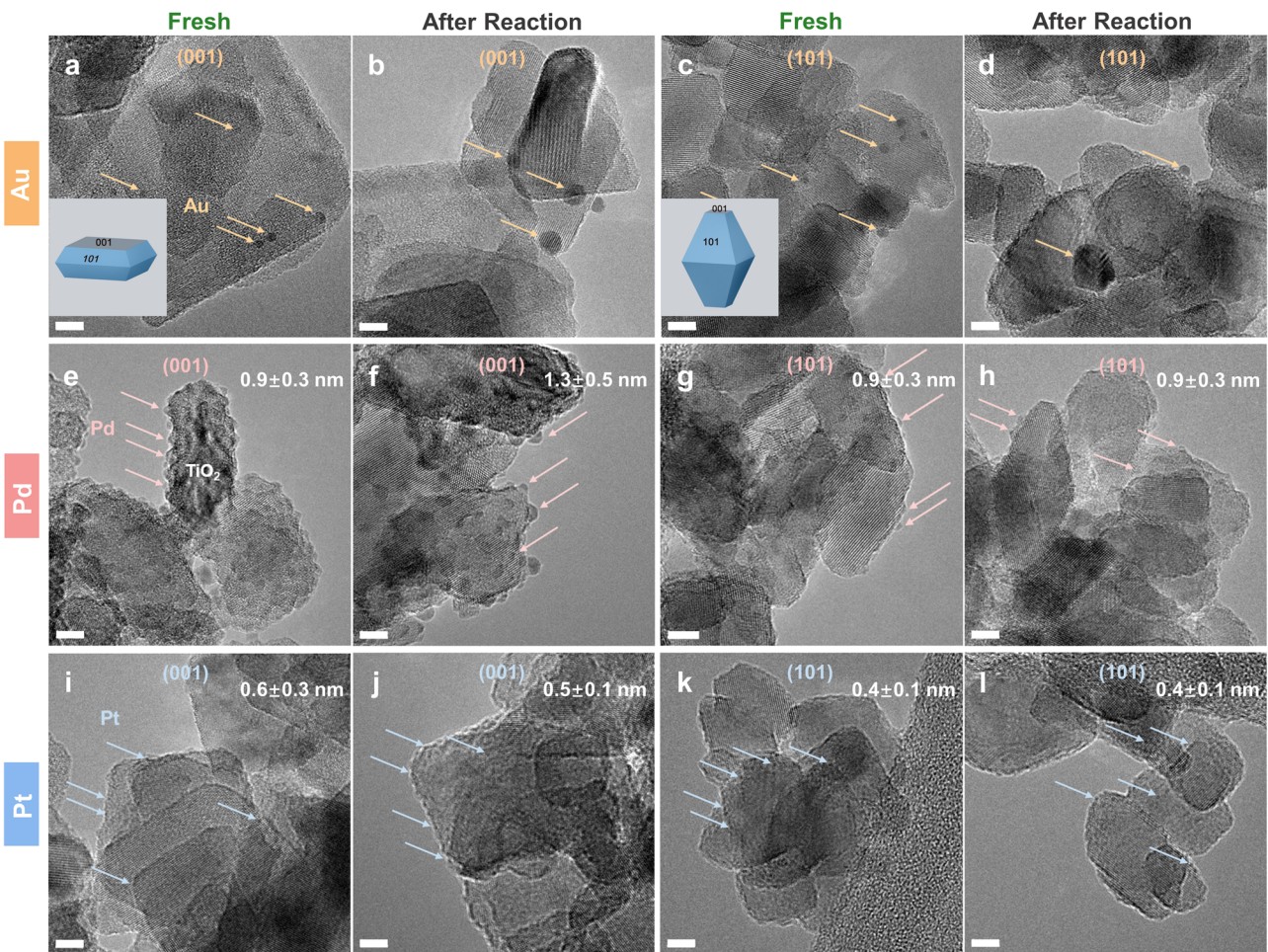

**Fig. 4 | HRTEM images of fresh and after photoreaction samples. a, b** Au/TiO$_2$-001. **c, d** Au/TiO$_2$-101. **e, f** Pd/TiO$_2$-001. **g, h** Pd/TiO$_2$-101. **i, j** Pt/TiO$_2$-001. **k, l** Pt/TiO$_2$-101. Scale bar: 5 nm. Metal clusters are indicated by arrows.

nanoclusters. The ranking of adsorption energies perfectly correlates with the amount of charge that the metal atoms transfer from their highly localized *d* orbitals to the anatase surface (Fig. 5b). The larger the amount of exchanged charge the stronger the metal adsorption on TiO$_2$ due to the appearance of attractive electrostatic forces on the surface and electronic hybridizations between *d* metal and *p* oxygen orbitals close to the Fermi energy level (Fig. 5c). These results suggest a decrease in metal-cluster stability according to the sequence Pt > Pd > Au when photoexcitation is absent.

To theoretically assess the stability of metallic clusters on TiO$_2$ surfaces under photoexcitation conditions, we adopted a DFT approach that allows for effective simulation of photoexcited carriers[52,53]. For the {001} facet, it was found that the stability ranking of nanoclusters was the same than determined under no illumination conditions, namely, Pt > Pd > Au. This result also can be understood in terms of the amount of *d* charge donated by the metal atoms to the anatase surface, which in all the cases increases upon the intensification of photoexcitation (Fig. 5d, e and Supplementary Fig. 16). In particular, Pt atoms transfer the largest amount of charge to their neighbouring Ti and O atoms, thus enhancing the reactivity of the surrounding anatase surface (Fig. 5f). These theoretical results, therefore, may explain in part the superior stability and H$_2$ production rate observed for Pt-based photocatalysts. It is worth noting that analogous results were obtained on Au nanoclusters adsorbed on the {001} facet (Supplementary Fig. 17).

As for the {101} facet, it was found that Au and Pd atoms can easily move on the TiO$_2$ surface when photoexcited (Fig. 5g, h), a result that

clearly indicates a tendency for Au and Pd agglomeration. Specifically, a photoinduced change in the preferred adsorption sites of Au and Pd atoms on the {101} surface spontaneously occurred during the geometry optimizations (i.e., from a preferred "hollow" adsorption site under mild photoexcitation conditions to a preferred "O-top" adsorption site under medium and intense photoexcitation conditions, Fig. 5g, h). In these two cases, the total amount of charge exchanged between the metal atoms and TiO$_2$ surface are minute (<0.1 $e^{-}$ as referred to in dark conditions) and the most notable electronic effects are intra-metallic $s \rightarrow p$ (Au/TiO$_2$-101) $d \rightarrow s$, $p$ (Pd/TiO$_2$-101) charge redistributions. Pt/TiO$_2$-101, on the other hand, behaves completely different from the analogous Au and Pd systems and very similarly to Pt/TiO$_2$-001, namely, photoexcitation does not induce any change on the preferred metal adsorption site and the amount of *d* charge donated to the anatase surface is substantial. Therefore, the stability and photoactivity of Pt/TiO$_2$-101 appear to be much superior to that of Au/TiO$_2$-101 and Pd/TiO$_2$-101, and similar to that of Pt/TiO$_2$-001. These theoretical DFT results are in consistent agreement with our experimental observations and reveal a central role played by photoinduced metal-anatase charge transfer on the stability and H$_2$ production rate of TiO$_2$-based photocatalysts.

Finally, we also performed first-principles DFT calculations to get insights into the possible causes of the differences in photocatalytic hydrogen activity experimentally observed for the Pt/TiO$_2$-101 and Pt/TiO$_2$-001 systems (Fig. 1 and Supplementary Fig. 18). Specifically, we followed the computational strategies explained in the previous works[54–57], since these allow to effectively estimate the performance of

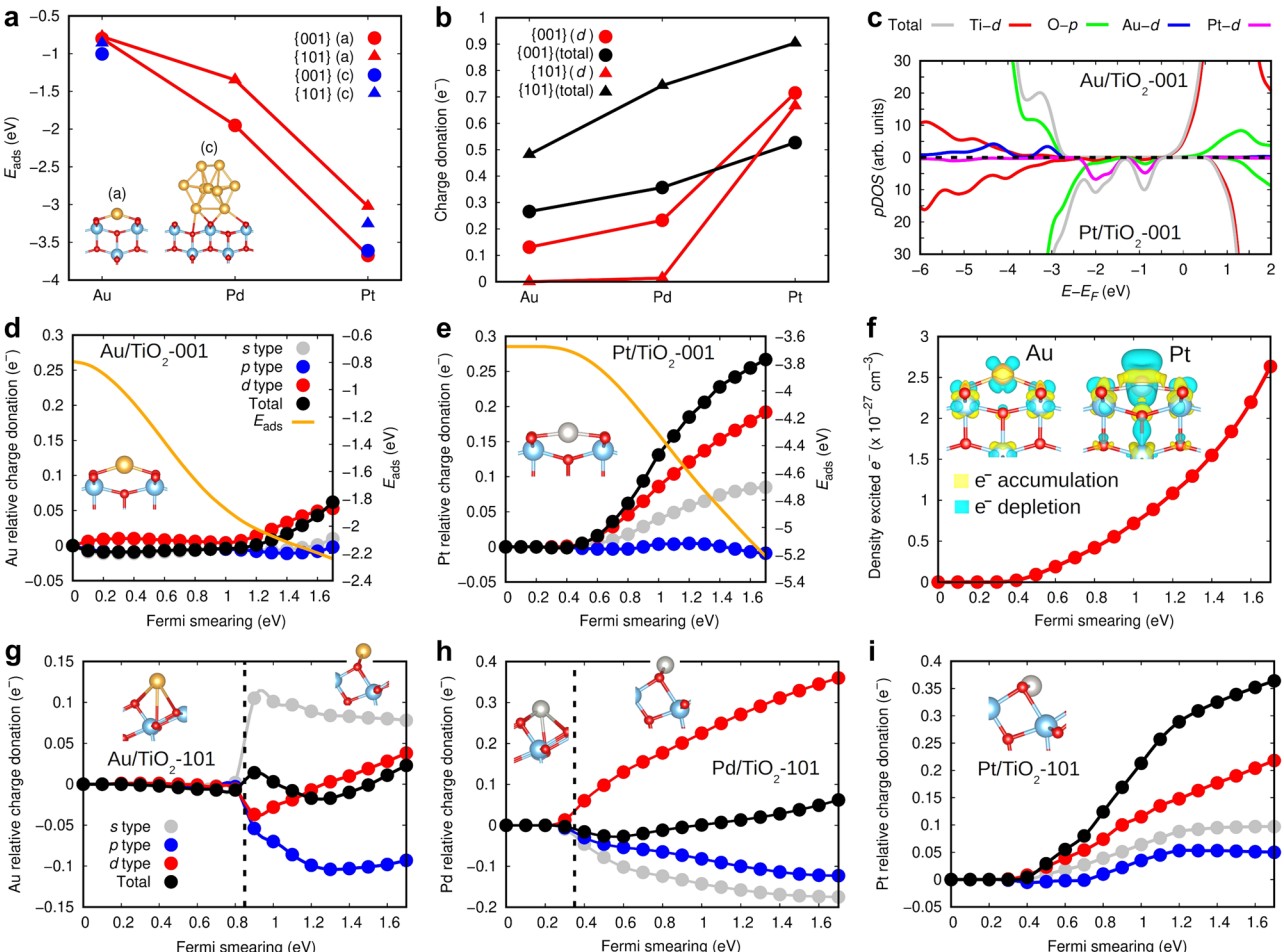

**Fig. 5 | First-principles simulations of noble metal blended anatase TiO₂ based on density functional theory. a** DFT adsorption energies; (a) stands for single atom and (c) for nanocluster. **b** Metal atoms charge donation; (d) stands for the highly localized electronic orbitals. **c** Partial density of electronic states for Au/TiO₂-001 and Pt/TiO₂-001. **d, e** Metal atoms charge donation under photoexcitation conditions on the {001} facet; results are referred to non-photoexcitation conditions and expressed as a function of the electronic Fermi smearing employed for promoting electrons from the valence to the conduction band (Methods). **f** Charge distribution difference analysis for Au and Pt atoms adsorbed on anatase TiO₂ and density of excited electrons expressed as a function of the electronic Fermi smearing. **g–i** Metal atoms charge donation under photoexcitation conditions on the {101} facet; results are referred to non-photoexcitation conditions. The dashed lines indicate photo-induced metal adsorption site transitions.

catalytic materials for the hydrogen evolution (HER) and oxygen evolution (OER) reactions through the calculation of adsorption free energies, $\Delta G$, of intermediate molecules involved in the oxidation of water and the reduction of hydrogen molecules (i.e., H*, O*, OH* and OOH*).

As it can be appreciated in the Supplementary Fig. 18a, in terms of HER performance the two Pt/TiO₂-101 and Pt/TiO₂-001 systems are very much alike, and in principle very efficient, since the estimated proton adsorption free energies, $\Delta G_{H*}$, are quite small in absolute value in both cases (i.e., 0.15–0.20 eV) and very similar in size to that of the archetypal HER catalyst Pt (111). On the other hand, in terms of OER activity (Supplementary Fig. 18b), Pt/TiO₂-101 turns out to be much superior than Pt/TiO₂-001 since the OER overpotential estimated for the former system is much smaller and comparable in size to those of archetypal OER catalysts like SrCoO₃, RuO₂ and PtO₂ (i.e., of the order 0.1 V). Therefore, based on these DFT simulations, the likely cause for the observed highest photocatalytic efficiency in H₂ production of Pt/TiO₂-101 (Fig. 1) is that in terms of the oxygen evolution reaction (OER) this system is much more efficient than Pt/TiO₂-001.

In summary, a simple mechanochemical strategy was employed to synthesize noble metal clusters (Au, Pd or Pt) supported on TiO₂

anatase plates or bipyramids preferentially exposing {001} or {101} facets, respectively. We systematically investigated the photocatalytic H₂ evolution for different metal clusters/anatase hybrids, and the ranking of H₂ production rate was Pt/TiO₂-101 ≫ Pd/TiO₂-001 > Pt/TiO₂-001 ~ Au/TiO₂-101 ≫ Au/TiO₂-001 > Pd/TiO₂-101 > TiO₂-001 ~ TiO₂-101. XPS analyses revealed that the {001} and {101} facets induce different oxidation states of metal co-catalysts during the milling process, thus affecting their photocatalytic performance. Furthermore, the long-term stability tests concluded a decreasing order of Pt/TiO₂-001 ~ Pt/TiO₂-101 > Pd/TiO₂-001 > Au/TiO₂-001 > Au/TiO₂-101. Specifically, the metal species behaved differently in terms of photocatalytic stability under same conditions, Pt/TiO₂ ≫ Pd/TiO₂ > Au/TiO₂, and the anatase plates were always more stable than the anatase bipyramids irrespective of the considered co-catalyst. HAADF-STEM and HRTEM images unambiguously depicted the phenomena of metal species aggregation before and after 20 h of photoreaction. Moreover, theoretical DFT simulations confirmed the crucial role of photoinduced metal-anatase charge transfers on the stability and H₂ production rate of TiO₂-based photocatalysts, which depends on the different metal/facet interfaces, in consistent agreement with the experimental observations.

## Methods

### Reagents

Gold (III) acetate (Alfa Aesar, 99.9%), Palladium (II) acetate (Acros Organics, 47.5% Pd), Platinum (II) acetylacetonate (Acros Organics, 98%) were used as metal precursors. Titanium (IV) fluoride ($TiF_4$, 99%), titanium (IV) chloride ($TiCl_4$, 99%), 1-octadecene (90%) (1-ODE), 1-octadecanol (1-ODOL, 97%). Oleylamine (OLAM, 70%) and oleic acid (OLAC, 90%) were obtained from Sigma-Aldrich. Absolute ethanol was obtained from Scharlau. Milli-Q water was routinely used.

### The preparation of TiO₂ photocatalysts

$TiO_2$ nanoshapes exhibiting (001) and (101) facets were prepared following a surfactant-assisted synthesis reported by Gordon et al.[13]. Highly monodisperse anatase tetragonal bipyramidal $TiO_2$ nanocrystals (NCs) exposing primarily {101} facets were produced using a mixture of $TiF_4$/$TiCl_4$ precursors in the presence of OLAM as a co-surfactant. Similarly, the use of 1-ODOL as a co-surfactant leads to $TiO_2$ nanoplates with a high percentage of {001} facets. $TiO_2$ NCs were prepared as follows: 30 mmol of co-surfactant (OLAM or 1-ODOL), 10.2 mL of 1-ODE, and 0.48 mL (1.5 mmol) of OLAC were degassed at 120 °C for 1 h. After, 1.5 mL of a mixed $TiF_4$/$TiCl_4$ stock solution 1:1 was added at 60 °C and the solution was quickly heated to 290 °C. After 10 min, 8 mL of the stock solution was pumped into the flask at 0.3 mL min$^{-1}$ using a syringe pump. Afterwards, the heating mantle was removed, and the flask was left to cool naturally to ambient temperature. After the synthesis, a mixture of 2-propanol and methanol is added to precipitate the NCs. Centrifugation at 4951 × $g$ for 10 min is used to recover them. This washing process was repeated twice. For the mixture of $TiF_4$/$TiCl_4$ precursors, $TiF_4$ and $TiCl_4$ stock solutions were mixed at equal volume in the glovebox. $TiF_4$ stock solution consists of 0.2 M $TiF_4$ and 1.0 M OLAC in 1-ODE. $TiCl_4$ stock solution consists of 0.2 M $TiCl_4$ and 1.0 M OLAC in 1-ODE. The $TiF_4$ stock solution is stirred on a hot plate set to 80 °C to promote the dissolution of $TiF_4$. Once dissolved, the $TiF_4$ stock solution is orange-brown, and the $TiCl_4$ stock solution is dark brown.

The $TiO_2$ anatase suspensions were evaporated overnight at room temperature. The obtained anatase powders were analysed through thermogravimetric measurements (TGA) to assess the temperature of complete mass loss of residual surfactants on the surface (Supplementary Fig. 19). Both materials were calcined in air at 500 °C for 1 h (2 °C min$^{-1}$) and the resulting materials were labelled as $TiO_2$-001 (for plates exhibiting predominantly {001} facets) and $TiO_2$-101 (for bipyramids exhibiting predominantly {101} facets). To facilitate an intuitive nomenclature of the samples that reflects the information of the primarily titania facet exposed, we named the samples as $TiO_2$-{hkl}. The XRD patterns (Supplementary Fig. 3) were used to confirm the formation of anatase without any impurities or rutile phase. TEM images (Supplementary Fig. 1a, b) unambiguously showed the synthesized shapes of anatase plates and bipyramids. At least one hundred $TiO_2$ nanoparticles in each sample were considered for estimating the particle size/shape distribution (Supplementary Fig. 2). On the basis of the obtained structural information, the percentages of preferentially exposed (101) facets in $TiO_2$ bipyramids and (001) facets in $TiO_2$ plates were estimated to be 91–97% and 50–76%, respectively. The detailed calculations are shown in Supplementary Fig. 20. The Fourier Transform (FT) images of fresh Pd/$TiO_2$ provide direct evidence of crystal planes of anatase {001} facets and {101} facets (Supplementary Figs. 1c and 1d, respectively).

### The preparation of M-TiO₂ photocatalysts

The mechanochemical preparation of the photocatalysts was performed in a high-energy RETSCH Mixer Mill MM200, using a 10 mL stainless-steel vial and one stainless-steel milling ball ($d$ = 15 mm, $m$ = 13.54 g), as described elsewhere[38,40]. Briefly, 0.0057 g of gold acetate, 0.0063 g of palladium acetate or 0.0061 g of platinum

acetylacetonate was mixed with 0.2943 g, 0.2937 g or 0.2939 g of $TiO_2$, respectively, leading to a ball to powder weight ratio (BPR) of 45 and metal loading of 1 wt%. Milling was performed at 15 Hz for 10 min. No signals of the precursors (acetate or acetylacetonate) were detected by either Raman spectroscopy or XPS (C 1s signal) in the samples prepared. The resulting metal/$TiO_2$ samples were denoted as Au/$TiO_2$-001, Au/$TiO_2$-101, Pd/$TiO_2$-001, Pd/$TiO_2$-101, Pt/$TiO_2$-001 and Pt/$TiO_2$-101. The obtained photocatalysts were used for the photocatalytic experiments without any further treatment. ICP-OES analyses (Supplementary Table 2) showed that all samples contained a similar metal loading of 0.9 ± 0.1 wt%, which is similar to the theoretical value of 1.0 wt%.

### Photocatalysts characterization

X-ray diffraction (XRD) data were collected on a PANalytical X'Pert diffractometer using a Cu Kα radiation source ($\lambda$ = 1.541 Å). In a typical experiment, the 2θ diffraction (Bragg) angles were measured by scanning the goniometer from 10° to 100°. The samples were prepared by centrifugation to precipitate the NCs. The supernatant was discarded, and samples were dried at room temperature. Peak positions were determined using the X'Pert HighScore program after baseline correction. High-angle annular dark field scanning transmission electron microscopy (HAADF-STEM) and high-resolution transmission electron microscopy (HRTEM) images were acquired on a FEI Tecnai G2 F20 instrument equipped with a field emission gun operating at an accelerating voltage of 200 kV. At least one hundred of Au, Pd and Pt particles in each photocatalyst were collected to estimate the size distribution. X-ray photoelectron spectroscopy (XPS) was conducted on a SPECS system. The apparatus was equipped with an XR-50 X-ray source (Al anode at 150 W), a PHOIBOS 150 EP hemispherical energy analyser and an MCD-9 detector. The spectra were corrected with the position of the adventitious carbon 1s signal at 284.8 eV. Thermogravimetric analyses (TGA) were performed on the TA Instrument with a Q50 system under nitrogen atmosphere (10 °C/min). The actual loading amount of Au, Pd and Pt was quantified by Inductively Coupled Plasma-Optical Emission Spectrometry (ICP-OES, Perkin Elmer Optima 3200RL). Surface area measurements were conducted on an automatic Micromeritics ASAP 2020 analyzer using $N_2$ adsorption isotherms and BET (Brunauer–Emmett–Teller) surface area analysis methods. Samples were degassed under vacuum at 200 °C for 4 h before adsorption measurements. The average pore diameter distributions were derived from the desorption branches of the isotherms based on Barrett-Joyner-Halenda (BJH) model.

### Photocatalytic hydrogen production

The photocatalytic hydrogen evolution reaction was conducted in a 40 mL tubular glass photoreactor at room temperature and atmospheric pressure under dynamic conditions. An UV lamp equipped with four LEDs emitting at 365 ± 5 nm (SACOPA, S.A.U.) was employed as the light source. An argon gas (Ar) stream (20 mL min$^{-1}$) was passed through a saturator (Dreschel bottle), which contained a liquid mixture of water ($H_2O$, 150 g) and ethanol (EtOH, 17 g). The resulting gaseous mixture of $H_2O$, EtOH and Ar was directly introduced into the photoreactor. 2.0 mg of each photocatalyst was dispersed in 0.5 mL absolute ethanol and treated by ultrasonication for 10 min to form a homogeneous suspension. Afterwards, the slurry was dropped onto a circular cellulose paper (from Albet, thickness 0.18 mm, area 2.54 cm$^2$) and dried at 50 °C for 1 h. The impregnated paper was placed upside down in the middle of the glass photoreactor with two separated tubular sections, along with an O-ring (Supplementary Fig. 6). The light source was positioned at the bottom of the photoreactor. By aligning a synthetic quartz glass cylindrical lens from the light source to the cellulose paper loaded with photocatalyst, 80.5 ± 0.5 mW cm$^{-2}$ of UV irradiance reached the sample. The cellulose paper was totally stable during drying/wetting, UV-exposure and gas permeation. The temperature of the photocatalyst under the UV irradiation was monitored

directly with a K-type thermocouple in contact with the sample. The gas hourly space velocity (GHSV) was 26,000 h⁻¹ and the contact time was 0.14 s. Prior to the photoreaction, the system was purged with Ar gas (20 mL min⁻¹) for 30 min. The products evolved from the outlet of photoreactor (hydrogen and acetaldehyde in equimolar amount, $C_2H_5OH \rightarrow H_2 + CH_3CHO$) were analysed on-line every 4 min with a micro-gas chromatograph (GC, Agilent 490)[58,59].

## First-principles simulation techniques

We performed spin-polarized first-principles calculations based on density functional theory (DFT) for anatase $TiO_2$ surfaces functionalised with Au, Pd and Pt co-catalysts. The PBE functional[60] was used as implemented in the VASP software package[61]. A "Hubbard-$U$" scheme[62] with $U = 3$ eV was employed for a better treatment of the localized Ti $d$ electronic orbitals (Hubbard-like corrections were not applied on the noble metal atoms). The value of the lattice parameters, however, were constrained to their corresponding experimental values of $a_0 = b_0 = 3.78$ Å and $c_0 = 9.51$ Å[63] since these are not correctly reproduced by the PBE + $U$ approach (Supplementary Methods and Supplementary Fig. 21). We used the "projector augmented wave" method to represent the ionic cores[64] by considering the following electrons as valence: Au 5$d$, and 6$s$; Pd 4$d$; Pt 5$d$ and 6$s$; Ti 3$d$, 4$s$, and 3$p$; and O 2$s$ and 2$p$. Wave functions were represented in a plane-wave basis truncated at 650 eV. For integrations within the first Brillouin zone, a Monkhorst-Pack k-point grid was employed with a density equivalent to $16 \times 16 \times 12$ for the anatase unit cell. Geometry relaxations were performed by using a conjugate-gradient algorithm that allowed for cell shape variations; the geometry relaxations were halted when the forces on the atoms fell all below 0.005 eVÅ⁻¹. By using these technical parameters we obtained zero-temperature energies converged to within 0.5 meV per formula unit.

All the DFT geometry relaxations in the present work were performed at the PBE + $U$ level with $U = 3$ eV and by constraining the size of the lattice vectors to their experimental values; the positions of the atoms, however, were allowed to fully relax in all the cases. Subsequently, the energy, charge density, and optoelectronic properties of anatase $TiO_2$ were estimated with the range-separated hybrid HSE06 exchange-correlation functional without performing further relaxations (Supplementary Methods and Supplementary Fig. 21).

The slab supercells were constructed as a $4 \times 4 \times 4$ repetition of the anatase unit cell and a vacuum region of 25 Å thickness was considered in all the simulations. Adsorption energies were calculated with the formula $E_{ads} = E_{M@anatase} - E_{anatase} - E_M$, where $E_{M@anatase}$ represents the energy of the blended noble metal-$TiO_2$ system, $E_{anatase}$ the energy of the anatase slab, and $E_M$ the energy of the isolated noble metal atom or cluster. To estimate accurate energy band gaps, density of electronic states and charge redistributions at affordable computational expense, we employed the hybrid HSE06 exchange-correlation functional[65] and adopted the equilibrium geometries determined at the PBE + $U$ level. In order to simulate photoexcitation effects in $TiO_2$-based photocatalysts, we employed an effective DFT approach, equivalent to those employed in previous works[52,53], that consists in constraining the partial occupancies of each electronic orbital by adjusting the width of the corresponding Fermi smearing. In particular, a Fermi smearing of σ (eV) in the constrained DFT calculations implies considering the electronic occupation function $f(E,\sigma) = 1/(\exp[(E - E_F)/\sigma]+1)$ instead of the usual step function $f(E) = 1$ for $E \leq E_F$ and $f(E) = 0$ for $E > E_F$, where $E_F$ represents the Fermi energy level.

Regarding the theoretical estimation of the performance of catalytic materials for the hydrogen evolution (HER) and oxygen evolution (OER) reactions, we followed the computational strategies explained in the previous works[54–57]. These strategies rely on the calculation of adsorption Gibbs free energies, $\Delta G$, of intermediate molecules involved in the oxidation of water and the reduction of hydrogen molecules (i.e., H*, O*, OH* and OOH*). The molecular adsorption

Gibbs free energies were estimated with the formula:

$$\triangle G_X = \triangle E_X + \triangle E_{ZPE} - T \triangle S$$

where $\Delta E_X$ corresponds to the molecular adsorption energy (that is, calculated under zero-temperature conditions) and $\Delta E_{ZPE}$ and $\Delta S$ are the difference in zero-point energy and entropy, respectively, between the molecule absorbed on the anatase surface and in the corresponding gas phase. The contribution from the anatase catalysts to both $\Delta E_{ZPE}$ and $\Delta S$ are very small and hence were neglected. The value of the $\Delta E_{ZPE}$ and $\Delta S$ terms were taken from the previous works[56,57].

## Data availability

The data that support the findings of this study have been included in the main text and Supplementary Information. The first-principles density functional theory calculations presented in this study have been deposited in a public repository at https://dataverse.csuc.cat/dataset.xhtml?persistentId=doi:10.34810/data756. All other relevant data supporting the findings of this study are available from the corresponding authors upon request.

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

## Acknowledgements

This work was supported by projects MICINN/FEDER PID2021-124572OB-C31 and GC 2021 SGR 01061. Y.C. (CSC No. 201806920042) acknowledges the China Scholarship Council for Ph.D. scholarship support. L.S. and C.C. are grateful to MICINN Ramon y Cajal program for individual fellowship grant agreements RYC2019-026704-I and RYC2018-024947-I, respectively. J.L. is a Serra Húnter Fellow and is grateful to ICREA Academia program. N.G.B. and V.P. acknowledge financial support from RTI2018-099965-B-I00, AEI/FEDER,UE and 2017-SGR-1431. ICN2 is supported by the Severo Ochoa program from Spanish MINECO (SEV-2017-0706) and is funded by the CERCA Programme/Generalitat de Catalunya. The authors thank E. Molins, I. Matas and M. Benito from ICMAB to kindly analyze BET areas.

## Author contributions

Y.C., L.S., J.O., N.G.B., V.F.P. and J.L. designed and performed the experimental studies. C.C. performed the computational studies. L.S. and J.L. supervised the work.

## Competing interests

The authors declare no competing interests.
