## [Peer Review File · Nature Communications]

Facet-engineered TiO₂ drives photocatalytic activity and stability of supported noble metal clusters during H₂ evolutionREVIEWER COMMENTS

Reviewer #1 (Remarks to the Author):

This work reports the overlooked aspect on the stability of the metal nanoparticles during the photoreaction. Their main new finding is that different TiO₂ facets give rise to different oxidation states in the metal clusters, the difference of which in each metal dictates photocatalytic activity. This work may potentially be publishable, but some issues need to be addressed before final decision is made.

1. Given the difference in the average sizes of metals loaded in each TiO₂ sample, the disparity observed in the stability of clusters against aggregation may have something to do with the size difference as well. While the authors claimed the nature of metal itself plays a significant role in the stability, the surface energy difference induced by the difference in size may have considerable effect on the stability of each metal cluster. The authors need to clarify this concern.
2. Although the same amount of each metal was loaded, the number of metal cluster in each sample may not be the same. This implies that the number of active sites was different for each sample. For more accurate comparison for photocatalytic activity, this aspect should be carefully considered. Otherwise, the conclusion they drew was not fully supported by the evidence they showed.
3. The resolution of TEM images of Pt-loaded TiO₂ in Figure 3 is so low that the presence of Pt clusters is not evidenced. More clear images are required to support their claim clearly.

Reviewer #2 (Remarks to the Author):

The presented data on the photocatalytic stability of faceted titania modified with three noble metals show some potential. However, they should be better discussed (including relevant literature), and further investigations (more detailed experiments) are necessary before getting the final conclusions. Authors should focus on the following aspects:

- 1) Data should be better discussed. For example, it is unknown why rate of hydrogen evolution is continuously increasing only for Pt/TiO₂-101 (even after initial time, i.e. > 10 min), whereas in the case of all other samples is not (Fig. 1a).
- 2) I am not sure if authors might conclude that the difference in activity between different samples is cause by the type of noble metals since much different sizes of nanoparticles have been used (Pt being much smaller – as suggested from Figures 3 and 4; Additionally, it is hardly to see any platinum deposits in Figure 3). Moreover, the content of noble metals seems to be much different – many Pd NPs and only few Pt NPs.
- 3) The names of samples could cause some confusion as it is obvious that 001 sample must also contain 101 facets. Additionally, 101 sample is not pure, also containing 001 facets.
- 4) The position of noble metals on different facets should be better investigated. Only one image (4 e) shows that metal is mainly deposited on 001 facet. Additionally, the obtained data should be better discussed (with relevant citations) as it has been commonly reported that noble metals are preferably deposited on (101) facet.
- 5) Moreover, authors should check relevant literature. The stability of noble metals on titania surfaces has already been reported, including also faceted titania. For example, the possible replacement of noble metals (from one to another facet) during photocatalytic reactions have been proposed.
- 6) I am not sure how theoretical calculations relate to the real photocatalytic data since UV light for H₂ evolution was used (for titania excitation). I think that similar conditions should be applied also for simulations.
- 7) I am not sure how the size of noble metals was estimated (supplementary Figure 4) since it is obvious that Pd formed much larger deposits than platinum (Figures 3 and 4).
- 8) What about the content of noble metals. It seems much different – Figure 3?
- 9) What about XPS data after long-term irradiation.
- 10) I am not sure what authors mean by “the transfer of photoexcited carriers from the metal species to the neighbouring Ti and O atoms”. It is well known (since A. J. Bard studies) that the transfer of

photogenerated electrons is from titania to noble metals (not opposite direction) under UV excitation (not considering plasmonic photocatalysis – but this could be possible only under vis radiation – so not the case shown here, i.e., H₂ evolution under UV).

Reviewer #3 (Remarks to the Author):

The work by Prof. Llorca and co-workers is an example of fundamental research aiming to shed light on the deactivation phenomenon of noble-metal-loaded TiO₂ nanoparticles by a combination of experiment and theory. Importantly for the community, the authors study the impact of metal-support interaction on co-catalyst sintering (during the photocatalytic process) and examine faceted TiO₂ powders that exhibit well-defined surfaces (which allows them to model the interactions using DFT). While I express my overall interest in the approach of this study, I have several critical points to the research design:

1. The first point is related to the choice of supporting TiO₂

A. To Fig S1: a, b: quality of the image is poor. The shape is not recognizable. Besides this, there seems a large size and shape distribution. c, d: the images rather emphasize my concern as no clear shape of the supporting TiO₂ could be identified (the same is often evident in Fig 3 and Fig 4 as unevenly shaped particles are apparent). Since these faceted TiO₂ samples are the basic supports used in this study to derive many conclusions about facet-dependent properties, the quality of the faceted TiO₂ powders needs to be examined in detail and its characteristics should be presented in a statistical way.

B. Next, since both {001} and {101} facets co-exist (!) on both the samples, it could be more correct to talk about the facet ratio rather than call the samples as “TiO₂-001” and “TiO₂-101”, which makes a wrong impression that one of the facets has very strong pre-dominance in the corresponding support and the rest of the particle surface can be neglected. This is far from the truth, especially for the TiO₂-001 sample, which is effectively a close to 50:50 mixture of both surfaces.

C. If I am right with point B, let me ask a question if (a) the mechano-chemical synthesis chosen in this work has any control over the facet-selective deposition of these co-catalysts OR if (b) your assumption is that the deposition of Au/Pt/Pd takes place on both of the co-existing facets in the extent that corresponds to their relative surface areas in the sample (e.g. approx. 50/50 for the TiO₂-001 sample)? If the second holds true (b), I would argue that your generic assumption that the HER performance of the two faceted TiO₂ samples under consideration (plates and bipyramids) can be explained by assuming that each of them has only one “dominant” facet is incorrect. Both facets will co-exist and other parameters (e.g. {001}/{101} ratio; absolute and relative BET surface area values; abundance of {001}/{101} interfaces that could exhibit outstanding catalytic profiles) may affect the observed HER trends.

2. The second point is related to the fact that the authors neglect other contributions to the photocatalyst instability derived from Figure 1b:

A. In lines 121-126, the authors arrive at a certain HER stability trend, however, this knowledge brings them to the sole conclusion that co-catalyst species rearrange (grow). I would like to argue against this narrow view prior to excluding other possibilities such as (a) TiO₂ (being a reducible metal oxide) can exhibit SMSI [10.1021/acscatal.7b00845], (b) oxidation state of the co-catalyst species (composition) can change upon illumination, (c) co-catalyst atoms may undergo sub-surface diffusion and thus affect the HER process in many different ways (10.1039/D1TA05561E). These and other possibilities shall be considered and checked as they as well will be highly dependent on both the nature of the surface (facet type) and the nature of the metal (co-catalyst type).

3. The third point is related to several important characterization data:

A. To XPS: XPS analyses were performed for the samples before catalysts, however – since the paper is focusing on the effect of facets on the stability and activity of the co-catalysts – it would be absolutely necessary to also provide XPS data after catalysis. I am quite sure that photocatalytic conditions (exposure to light) also result in further reduction of Au/Pd/Pt species resulting in a higher proportion of metallic co-catalysts. Importantly, this process (photoreduction) can also be facet-dependent and will also likely contribute to the observed activity trends.

B. To XPS discussion (line 136-154): please clarify what you mean by oxidation of Pt/Pd/Au. Do you expect that some of the co-catalyst nanoparticles (e.g. those sitting on a specific facet) are in oxidation state 0 (metallic form), while others are +2/+4 (oxide form). Or do you mean that all and each of the co-catalyst nanoparticles is made of a mixture of oxidation states? In the second case, do you assume that oxidized metals (+2/+4) are at the TiO₂/NP interface or on the NP/air interface? This is important and may need to be considered in both the HER results interpretation and DFT model justification.

C. To STEM/HRTEM: all the images presented in the paper are taken from the samples after 1-hour HER runs, however, according to HER profiles in Figure 1, no (noticeable) deactivation has yet taken place. In fact, the HER deactivation gets prominent only after several hours of light exposure, which raises many questions to the relevance of particle growth to the HER performance.

D. To DFT: most of the conclusions drawn in this section are directly related to the fact that Pt has the highest adsorption energy on TiO₂, which is related to its strongest electronic interaction and results in its highest stability against atomic movement (i.e. sintering) compared to Pd and Au. I do not find the models to be quite appropriate to account for the complexity of the photocatalytic systems, I also do not see the authors considering any experimental data input to shape the calculations/models.

Besides these, let me convey a few other less important points, that nevertheless would need clarifications:

4. How sure the authors are that TiO₂ (its phase composition and morphology) did not change upon mechano-activation? At least XRD and BET data should be provided.

On the same line, (sorry for picking on the data), after looking at Fig S9: could it be that the TiO₂-001 pattern also features a tiny rutile peak around 27 degrees? I am asking because 500 C treatment could be high enough to facilitate at least a partial A to R transition which needs to be considered.

5. While the authors make a strong point in the introduction that the choice of the conditions (surface termination, type of co-catalyst, type of reaction) make a strong impact on the photocatalytic performance, their experiments are focused on quite a specific combination of using gas-phase HER and water-EtOH vapor as a reactant. I would suggest that the conclusions derived in this work will likely not apply to a much more typical liquid-media suspension-based photocatalytic HER experiments, and I would thus not agree with the statement in line 89 that they “can be generalized to other types of reactions and probably also to other supports (e.g., rutile TiO₂ and CeO₂).” This, of course, limits the relevance of this research work.

6. I am not satisfied with the discussion of the previous research in the introduction. Starting from line 41, the authors jump into the statement that {001} is more efficient than {001}. In a few sentences, they highlight that the choice of the reaction is very important as well as the use of co-catalyst, so why did not they clarify these parameters in their first statement?

I generally agree that a lot of factors have to be considered when making such statements, but the introduction makes the impression that nothing has been clarified at all in the literature. In fact, the

authors claim in the abstract that "systematic studies on" the "stability during the photoreaction" ... "are absent for metal clusters supported on TiO₂". This is incorrect and many literature pieces (also cited in the paper) do shed light on the issues.

7. To DFT: how do the DFT models (cluster on TiO₂ and single-atom on TiO₂) correspond to the XPS findings? Do authors think that considering experimental data (oxidation state distribution) is important in order to construct reliable DFT models?

8. To DFT, lines 215-217: I do not follow how is this conclusion evident. What is the method to evaluate this? What are the results for Pt? I assume that this result is also obvious from the previous data as Pt was shown to stick to TiO₂ most strongly. This exactly would correspond to the lowest probability of Pt moving from one place to another. Please clarify your method of arriving at this conclusion possibly in the experimental section.

Line 56: "it is well known" requires a reference.

Line 60: "...many previous studies.." requires a reference or a set of references

Line 72: it should rather read "thiolate-protection, and polymer stabilization"

Line 97: I find that the storyline (the Results section starts from HER data) is missing the basic sample characterization details, which I would suggest the authors to highlight briefly at the beginning of the section. Basically, mentioning Fig S8, S9 and the XRF data (not found). I would also strongly advise adding BET values (relevant to my comment 1)

Line 98: I would call your "4 (recyclability)" experiment as a "4x1h light on-off cycle experiment". Recyclability is often associated with the reuse (recovery) of the solid-state catalyst and the notion can thus be misleading.

Manuscript "Facet-engineered TiO₂ drives photocatalytic activity and stability of supported noble metal clusters during H₂ evolution"

NCOMMS-22-42901-T

REPLY TO THE COMMENTS RAISED BY THE REVIEWERS

Reviewer #1

This work reports the overlooked aspect on the stability of the metal nanoparticles during the photoreaction. Their main new finding is that different TiO₂ facets give rise to different oxidation states in the metal clusters, the difference of which in each metal dictates photocatalytic activity. This work may potentially be publishable, but some issues need to be addressed before final decision is made.

ANSWER: We are grateful to the positive comments of the reviewer. We would like to add that, in addition to the main finding described by the reviewer, we also report a systematic study of the stability of different metals supported on different titania facets during the hydrogen evolution reaction, which, in our opinion, represents a substantial contribution to the existing literature and a valuable piece of information for the community working on the photoproduction of hydrogen, since it highlights the role of different metals and titania facets on photostability.

1. Given the difference in the average sizes of metals loaded in each TiO₂ sample, the disparity observed in the stability of clusters against aggregation may have something to do with the size difference as well. While the authors claimed the nature of metal itself plays a significant role in the stability, the surface energy difference induced by the difference in size may have considerable effect on the stability of each metal cluster. The authors need to clarify this concern.

ANSWER: This is a very important and interesting point to discuss since it could be possible that the initial cluster size could be related to the cluster stability (cluster growth), motivated by differences in surface energy. In our study, however, we would like to argue that: (i) there is no trend between the initial size of the clusters and the sizes after the photocatalytic experiments. This is particularly evident when comparing the Au cluster growth between the Au/TiO₂-001 and Au/TiO₂-101 samples, where the initial size of the Au clusters is the same for both samples (1.5-1.6 nm) whereas after the photocatalytic experiment the mean Au cluster size is totally different, 1.8 and 3.3 nm, respectively (see Figure 3). This unambiguously demonstrates that the different facets of titania play an active role on the growth of the metal clusters deposited on them, which is not related to any size effect. (ii) When comparing Au and Pd clusters, the mean initial cluster size for all of them deposited on the different titania facets is again virtually

identical and within the experimental measurement uncertainty (1.3-1.6 nm, see Figure 3). However, the growth experimented by the Pd clusters is clearly lower than that of the Au clusters, irrespective of the titania facet where the clusters are deposited. This is a clear indication that the growth of the clusters after the photocatalytic experiment is not related to cluster size, but to the nature of the metal considered. (iii) In the case of Pt clusters, it is true that the initial cluster size is lower with respect to Au and Pd clusters, but in this case the Pt clusters do not growth at all. Considering that the cluster surface energy increases as the cluster size decreases, the largest stability exhibited by Pt clusters with respect to Au and Pd clusters clearly indicates that stability under photocatalytic conditions is not related to the initial size of the metal clusters, at least in the size range considered in our work and under the photoreaction conditions employed. Therefore, we can safely conclude that the differences observed in the stability of clusters against aggregation are not related to differences in the initial cluster size.

2. Although the same amount of each metal was loaded, the number of metal cluster in each sample may not be the same. This implies that the number of active sites was different for each sample. For more accurate comparison for photocatalytic activity, this aspect should be carefully considered. Otherwise, the conclusion they drew was not fully supported by the evidence they showed.

ANSWER: For a given metal loading, the number of individual metal clusters depends on the cluster size. However, and in a different way than it usually occurs in thermal catalysis, the number of active sites in photocatalysis not always correlates with cluster size, as the mechanism of accepting photogenerated charge carriers is strongly dependent on the electronic properties of the co-catalyst, which can be strongly influenced by the metallicity of the metal cluster, which in turn could be correlated to cluster size. In other words, the identification of active sites in photocatalytic processes is actually a hot topic in the community and much work still needs to be done. Obviously, this falls beyond our work. However, we would like to recall that our results do not show a trend between photoactivity and cluster size when the different metal clusters are considered for a given titania facet. For instance, as shown in Figure 1, the rate of hydrogen formation on TiO₂-001 is Pd/TiO₂-001 > Pt/TiO₂-001 > Au/TiO₂-001 whereas the metal cluster size is Au/TiO₂-001 ~ Pd/TiO₂-001 > Pt/TiO₂-001, and the rate of hydrogen formation on TiO₂-101 is Pt/TiO₂-101 > Au/TiO₂-101 > Pd/TiO₂-101 whereas the metal cluster size is Au/TiO₂-101 ≥ Pd/TiO₂-101 > Pt/TiO₂-101. Therefore, we can safely conclude that photoactivity does not correlate with cluster size.

3. The resolution of TEM images of Pt-loaded TiO₂ in Figure 3 is so low that the presence of Pt clusters is not evidenced. More clear images are required to

support their claim clearly.

ANSWER: The referee is right, and we apologise for the inconvenience. For the sake of clarity, we have added in the supplementary information enlarged HRTEM images recorded on Pt-loaded titania to better visualise the Pt clusters (new Supplementary Figure 8 and 9).

Reviewer #2

The presented data on the photocatalytic stability of faceted titania modified with three noble metals show some potential. However, they should be better discussed (including relevant literature), and further investigations (more detailed experiments) are necessary before getting the final conclusions. Authors should focus on the following aspects:

ANSWER: We are grateful to the reviewer for his/her appreciation of our work as well as for the detailed observations and questions raised. We have revised again the existing literature to accurately reflect the work already reported and to enrich our discussion. Also, we have performed additional experiments when required.

1) Data should be better discussed. For example, it is unknown why rate of hydrogen evolution is continuously increasing only for Pt/TiO₂-101 (even after initial time, i.e. > 10 min), whereas in the case of all other samples is not (Fig. 1a).

ANSWER: This is a very interesting point. In fact, for both Pt/TiO₂-101 and Pt/TiO₂-001 a progressive improvement of the hydrogen evolution rate is observed (Figure 1). This phenomenon for gas-phase hydrogen evolution from water-ethanol has been recently described in the literature by Dessal et al. (Journal of Catalysis 2019, 375, 155-163) and later studied by operando X-ray absorption spectroscopy-mass spectrometry by Piccolo et al. (ACS Catalysis 2020, 10, 12696-12705). It has been demonstrated that Pt activation is due to a progressive Pt reduction under photoreaction conditions. In order to address the reader properly, we have added a sentence in the revised manuscript regarding the activation of Pt-loaded titania samples under photoreaction (page 5). The related works of Dessal et al. and Piccolo et al. have been properly cited as well.

2) I am not sure if authors might conclude that the difference in activity between different samples is caused by the type of noble metals since much different sizes of nanoparticles have been used (Pt being much smaller – as suggested from Figures 3 and 4; Additionally, it is hardly to see any platinum deposits in Figure 3). Moreover, the content of noble metals seems to be much different – many Pd NPs and only few Pt NPs.

ANSWER: As the reviewer says, the mean Pt cluster size is smaller than that of Pd and Au clusters, but the mean sizes of Pd and Au clusters are virtually indistinguishable within the experimental accuracy (see Figures 3 and 4) and they certainly show different photocatalytic activity. In this sense, we can state that our results do not show a trend between photoactivity and cluster size when the different metal clusters are considered for a given titania facet. For instance, as shown in Figure 1, the rate of hydrogen formation on TiO₂-001 is Pd/TiO₂-001 > Pt/TiO₂-001 > Au/TiO₂-001 whereas the metal cluster size is Au/TiO₂-001 ~ Pd/TiO₂-001 > Pt/TiO₂-001, and the rate of hydrogen formation on TiO₂-101 is Pt/TiO₂-101 > Au/TiO₂-101 > Pd/TiO₂-101 whereas the metal cluster size is Au/TiO₂-101 ≥ Pd/TiO₂-101 > Pt/TiO₂-101. Therefore, we can safely conclude that photoactivity does not correlate with cluster size. Regarding the presence of Pt deposits, the referee is right and we apologise for the inconvenience. We have added in the supplementary information enlarged HRTEM images recorded on Pt-loaded titania to better visualise the Pt clusters (new Supplementary Figures S8 and S9). Finally, the content of noble metals has been checked by ICP and all samples contain the nominal 1 wt% loading (0.9 ± 0.1 wt%). We have added a sentence in this regard in the “methods” section of the revised manuscript (page 14). It should be stressed out that the dry mechanochemical preparation method used allows obtaining very homogeneous materials with elemental contents virtually identical to the nominal values (see, for instance Applied Catalysis B: Environmental 2022, 309, 121275).

3) The names of samples could cause some confusion as it is obvious that 001 sample must also contain 101 facets. Additionally, 101 sample is not pure, also containing 001 facets.

ANSWER: The reviewer is totally right; our titania samples contain both {001} and {101} facets. The point is that, as explained in the “methods” section (Photocatalyst Preparation), by using a surfactant-assisted route, bipyramidal nanocrystals exposing primarily {101} facets and nanoplates exposing primarily {001} facets have been synthesized. These nanoparticles have been used as model titania facets, which obviously do not constitute 100% of the facets exposed. To facilitate an intuitive nomenclature of the samples that reflects the information of the primarily titania facet exposed, we would like to maintain the names of the samples as M/TiO₂-{hkl}. Nevertheless, and according to the comment raised by the reviewer, we have modified the abstract and added a sentence in the revised manuscript (page 13, and Supplementary Information page 5) to avoid any confusion.

4) The position of noble metals on different facets should be better investigated. Only one image (4e) shows that metal is mainly deposited on 001 facet. Additionally, the obtained data should be better discussed (with relevant citations)

as it has been commonly reported that noble metals are preferably deposited on (101) facet.

ANSWER: This is certainly an interesting issue. From our TEM study, we can unambiguously conclude that in our samples there is no differential deposition of metal clusters on different titania facets, neither in the fresh samples nor in the post-reacted ones. This is explained based on the dry mechanochemical preparation method used, which, in contrast to wet chemistry methods, does not yield any preferential deposition because the samples are prepared by direct, high-energy milling. In the literature, preferential deposition of metal clusters is usually accomplished by using the wet photodeposition method (photoreduction), that allows the formation of metal clusters on specific facets due to the reduction of the metal precursor by the photoexcited electrons enriched on the particular facet (see, for instance, Zhang et al., *Journal of Catalysis* 2016, 337, 36-44, Wei et al., *Applied Catalysis B: Environmental* 237 (2018) 574–587). We have added an explanatory sentence in this regard in the revised version of the manuscript (page 8).

5) Moreover, authors should check relevant literature. The stability of noble metals on titania surfaces has already been reported, including also faceted titania. For example, the possible replacement of noble metals (from one to another facet) during photocatalytic reactions have been proposed.

ANSWER: Following the suggestion of the reviewer, we have included additional relevant literature in the revised version of the manuscript that we missed in the original manuscript (new references 50 and 51) to enrich our discussion related to the stability of noble metals on titania facets.

6) I am not sure how theoretical calculations relate to the real photocatalytic data since UV light for H₂ evolution was used (for titania excitation). I think that similar conditions should be applied also for simulations.

ANSWER: The label “Photoexcitation” in Figures 5d,e,g-i, certainly cannot be directly compared to the energy of the radiation employed in the photocatalytic experiments. As it appears explained in the caption of Figure 5 and the Methods section, “The photoexcitation energy corresponds to the width of the electronic Fermi smearing” employed for constraining the partial occupancies of the electronic orbitals in the DFT simulations (or, in other words, for transferring electrons from the valence to the conduction band in an effective manner and evade electron-hole recombination). We recall here that a Fermi smearing of σ (eV) in the constrained DFT calculations implies considering the electronic occupation function $f(E,\sigma)=1/(\exp[(E-E_F)/\sigma]+1)$, instead of the usual step function $f(E)=1$ for $E \leq E_F$ and $f(E)=0$ for $E > E_F$, where E_F represents the Fermi energy level.

Explicitly simulating the interactions between light and matter in materials with DFT methods is an extremely complex and laborious task (due to the presence of, for instance, photon-phonon interactions and charge recombination processes involving non-equilibrium phenomena), which is out of the scope of the present work. To mimic the results of electronic photoexcitation in X-TiO₂ as induced by UV radiation, we employed instead an effective approach that consists in constraining the electronic occupation through Fermi smearing; thus, we can mimic the results of photoexcitation (i.e., promotion of electrons from the valence to the conduction band) without the need of explicitly simulating photoexcitation. In the present case, the Fermi energy level is positioned close to the conduction band (see Fig.5c), hence smaller values of the Fermi smearing as compared to those of UV radiation are sufficient to promote electrons from the valence to the conduction band. Obviously then, a “one-to-one” correspondence between the Fermi smearing employed in the effective DFT calculations and the light frequency employed in the experiments is not possible. One quantity though that can be extracted from our effective DFT simulations, and which is more interpretable in terms of measurable quantities, is the density of excited electrons (i.e., the number of electrons promoted from the valence to the conduction band divided by the volume of the crystal). In the revised version of our article, we have applied two major modifications on Figure 5 for avoiding possible misunderstandings. First, the label “Photoexcitation (eV)” has been replaced by “Fermi smearing (eV)”. And second, we have specified the value of the density of excited electrons as a function of the employed Fermi smearing. In addition, we have also added some supplementary explanations about the employed effective DFT approach in the Methods section.

7) I am not sure how the size of noble metals was estimated (supplementary Figure 4) since it is obvious that Pd formed much larger deposits than platinum (Figures 3 and 4).

ANSWER: The determination of metal cluster sizes is not straightforward. After more than 25 years our impression is that this is still a pending issue of the community working on catalysis and photocatalysis. There are different characterization techniques to directly determine the size of nanoparticles of a given size with high statistics, but subnanometer to 2 nm sized supported clusters are difficult to measure with accuracy, mainly because transmission electron microscopy techniques are required, which are limited by statistics. When such small metal entities possess magnetic properties, a precise determination is possible via magnetic measurements, but this is not our case, where the size of Au, Pd and Pt clusters can only be directly determined with accuracy with transmission electron microscopy, although the measurement is limited to various hundreds of particles only. We have used HRTEM in addition to HAADF-STEM

because, with our instrument, subnanometer clusters escape detection with HAADF-STEM (noble metals are easier to identify by HAADF-STEM given the difference in atomic weight with respect to the titania support, but we do not have access to an aberration-corrected instrument). For that reason, the mean cluster size of Pd and Pt determined by HAADF-STEM can be biased toward larger values with respect to HRTEM measurements. Honestly, given these uncertainties, we have chosen to show both HAADF-STEM and HRTEM data. Nevertheless, beyond the exact measurement of the mean metal cluster size values, the most important observation here is related to the relative growth of the different clusters on the different titania facets during the photocatalytic reaction, and we have convincingly demonstrated the different behaviour of the different noble metals as well as the role played by the titania facet.

8) What about the content of noble metals. It seems much different – Figure 3?

ANSWER: As explained in the answer to the question 2 raised by the reviewer, the content of noble metals has been checked by ICP and all samples contain a metal loading of 0.9 ± 0.1 wt%. As explained above, we have added a sentence in this regard in the “methods” section of the revised manuscript (page 14).

9) What about XPS data after long-term irradiation.

ANSWER: This is an interesting point that certainly complements our discussion and we have collected the XPS of the post-reacted samples that underwent deactivation (Au and Pd, see new Figure S7). We have found that Au in both Au/TiO₂-001 and Au/TiO₂-101 was mainly reduced under the long-term UV irradiation. The metallic Au species (Au⁰) in Au/TiO₂-001 and Au/TiO₂-101 samples increased from 69.9% and 51.5% to 80.9% and 69.4%, respectively. Pd/TiO₂-001 exhibited a similar oxidation state of Pd components before and after photoreaction, whereas Pd/TiO₂-101 also underwent a reduction process. The change in the oxidation states of metal clusters during the photoreaction can be explained by the reduction of the metal species induced by the photogenerated electrons under the ethanol/water and UV irradiation condition, which also has been observed in previous studies [Lee et al. Nat. Mater. 18, 620–626 (2019); Piccolo et al. ACS Catal. 2020, 10, 12696–12705]. We have added the corresponding discussion in the revised manuscript (page 7).

10) I am not sure what authors mean by “the transfer of photoexcited carriers from the metal species to the neighbouring Ti and O atoms”. It is well known (since A. J. Bard studies) that the transfer of photogenerated electrons is from titania to noble metals (not opposite direction) under UV excitation (not considering plasmonic photocatalysis – but this could be possible only under vis radiation – so not the case shown here, i.e., H₂ evolution under UV).

ANSWER: The reviewer is raising an interesting question that is currently a hot discussion in the heterogeneous photocatalytic community. Whereas the pioneering studies of A. J. Bard suggested that “the transfer of photogenerated electrons is from titania to noble metals (not opposite direction) under UV excitation (not considering plasmonic photocatalysis)”, recent studies focused on the principles of band bending and the different work function values of noble metals (the work function of a material can be defined as the energy required to extract an electron from the surface of the material, and can be calculated considering the position of the Fermi level respect the vacuum) in order to tune the charge carrier behaviour (see, for instance, relevant reviews Zhang and Yates, Chem. Rev. 2012, 112, 5520-5551 and Yanagi et al., ACS Energy Lett. 2022, 7, 432-452, among others). Actually, the idea of manufacturing photocatalytic materials (semiconductors) that include cocatalysts acting as selective contacts for one type of charge carriers is well established in photovoltaics, where the junction of a light absorbent material (semiconductor) sandwiched between two different materials induces the generation of hole transport layers (HTL) and electron transport layers (ETL) that act as selective contacts (see for instance Puigdollers et al., Energy Harvesting & Storage 2022, 61-95, ISBN 978-981-19-4526-7). Regarding photocatalytic applications, a number of DFT calculations and experimental studies show the band bending of Au/TiO₂ and the Au effect in hole transport (see for instance Zhang et al., J. Phys. Chem. C, 2011, 115, 23848-23853). Additionally, our DFT calculations are in agreement with those abovementioned studies, suggesting that the work functions of Au, Pd and Pt will induce those noble metals to act as HTL when are anchored to TiO₂ under UV illumination (non-equilibrium conditions).

Reviewer #3

The work by Prof. Llorca and co-workers is an example of fundamental research aiming to shed light on the deactivation phenomenon of noble-metal-loaded TiO₂ nanoparticles by a combination of experiment and theory. Importantly for the community, the authors study the impact of metal-support interaction on cocatalyst sintering (during the photocatalytic process) and examine faceted TiO₂ powders that exhibit well-defined surfaces (which allows them to model the interactions using DFT). While I express my overall interest in the approach of this study, I have several critical points to the research design:

ANSWER: We are very grateful to the positive comments of the reviewer and for his/her appreciation of our work as well as for the detailed observations and questions raised.

1. The first point is related to the choice of supporting TiO₂

A. To Fig S1: a, b: quality of the image is poor. The shape is not recognizable. Besides this, there seems a large size and shape distribution. c, d: the images rather emphasize my concern as no clear shape of the supporting TiO₂ could be identified (the same is often evident in Fig 3 and Fig 4 as unevenly shaped particles are apparent). Since these faceted TiO₂ samples are the basic supports used in this study to derive many conclusions about facet-dependent properties, the quality of the faceted TiO₂ powders needs to be examined in detail and its characteristics should be presented in a statistical way.

ANSWER: Following the reviewer's comment, we have replaced the original images with new TEM images (see Supplementary Fig. 1a and 1b), which show that titania plates and bipyramids were successfully synthesized. The size distribution of TiO₂ nanoparticles has been estimated based on several hundred individual nanoparticles (Supplementary Fig. 2). The average side length and thickness of TiO₂-001 are 31.1 and 9.8 nm, respectively. The average side length and thickness of TiO₂-101 are 12.3 and 23.6 nm, respectively. The corresponding description has been added in the revised manuscript (page 4 and 13).

B. Next, since both {001} and {101} facets co-exist (!) on both the samples, it could be more correct to talk about the facet ratio rather than call the samples as "TiO₂-001" and "TiO₂-101", which makes a wrong impression that one of the facets has very strong pre-dominance in the corresponding support and the rest of the particle surface can be neglected. This is far from the truth, especially for the TiO₂-001 sample, which is effectively a close to 50:50 mixture of both surfaces.

ANSWER: The reviewer is totally right; our titania samples contain both {001} and {101} facets. The difference is that titania bipyramids expose preferentially {101} facets and titania plates expose preferentially {001} facets. We have performed a detailed statistical analysis (Supplementary Fig. 16) and have found that the percentage of preferentially exposed {101} facets in the TiO₂ bipyramids is estimated to be 91%–97%, and for titania plates the percentage of preferentially exposed {001} facets is estimated to be 50%–76%. Even if the titania nanoparticles used contain different amount of {001} and {101} facets, we would like to maintain the names of the samples as M/TiO₂-{hkl} to facilitate an intuitive nomenclature of the samples that reflects the information of the primarily titania facet exposed. For the sake of clarity, we have revised the text thoroughly to avoid any confusion in this respect (abstract and page 13 in the revised manuscript, and page 5 in the Supplementary Information).

C. If I am right with point B, let me ask a question if (a) the mechano-chemical synthesis chosen in this work has any control over the facet-selective deposition of these co-catalysts OR if (b) your assumption is that the deposition of Au/Pt/Pd

takes place on both of the co-existing facets in the extent that corresponds to their relative surface areas in the sample (e.g. approx. 50/50 for the TiO₂-001 sample)? If the second holds true (b), I would argue that your generic assumption that the HER performance of the two faceted TiO₂ samples under consideration (plates and bipyramids) can be explained by assuming that each of them has only one “dominant” facet is incorrect. Both facets will co-exist and other parameters (e.g. {001}/{101} ratio; absolute and relative BET surface area values; abundance of {001}/{101} interfaces that could exhibit outstanding catalytic profiles) may affect the observed HER trends.

ANSWER: From our TEM study, we can unambiguously conclude that in our samples there is no differential deposition of metal clusters on different titania facets, neither in the fresh samples nor in the post-reacted ones. This is explained based on the dry mechanochemical preparation method used, which, in contrast to wet chemistry methods, does not yield any preferential deposition because the samples are prepared by direct, high-energy milling. In the literature, preferential deposition of metal clusters is usually accomplished by using the wet photodeposition method (photoreduction), that allows the formation of metal clusters on specific facets due to the reduction of the metal precursor by the photoexcited electrons enriched on the particular facet (see, for instance, Zhang et al., *Journal of Catalysis* 2016, 337, 36-44). We have added an explanatory sentence in this regard in the revised version of the manuscript (page 8). Since titania bipyramids expose preferentially {101} facets and titania plates expose preferentially {001} facets, by comparing the behaviour of both samples loaded with the same metal clusters, we can get insight into the effect of each facet exposed on photocatalyst activity and stability. It is true that a direct and straightforward comparison may be delicate, and for that reason we have not attempted to perform a quantitative evaluation. Also, we agree with the reviewer that we can not rule out the possibility that {001}/{101} interfaces could exhibit catalytic properties, as suggested, for instance, by Yu et al. for the photocatalytic reduction of carbon dioxide (*J. Am. Chem. Soc.* 2014, 136, 8839-8842), but there is a general agreement on the fact that edges and defects in titania usually act as recombination centres for electron-hole pairs, thus lowering the photocatalytic activity. We fully acknowledge the reviewer for this interesting observation.

2. The second point is related to the fact that the authors neglect other contributions to the photocatalyst instability derived from Figure 1b:

A. In lines 121-126, the authors arrive at a certain HER stability trend, however, this knowledge brings them to the sole conclusion that co-catalyst species rearrange (grow). I would like to argue against this narrow view prior to excluding other possibilities such as (a) TiO₂ (being a reducible metal oxide) can exhibit SMSI [10.1021/acscatal.7b00845], (b) oxidation state of the co-catalyst species (composition) can change upon illumination, (c) co-catalyst atoms may undergo

sub-surface diffusion and thus affect the HER process in many different ways (10.1039/D1TA05561E). These and other possibilities shall be considered and checked as they as well will be highly dependent on both the nature of the surface (facet type) and the nature of the metal (co-catalyst type).

ANSWER: The point raised by the reviewer is again stimulating. We would like to stress out that: (i) SMSI is unlikely to occur in our case because we have used the photocatalysts as prepared and without any reduction treatment and the UV irradiance used in the photocatalytic experiments is really low, making the formation of oxygen vacancies and SMSI unlikely (see, for comparison Haselmann and Eder, ACS Catalysis 2017, 7, 4668-4675). Also, we have not seen any thin shell on metal nanoparticles after reaction by HRTEM that could be ascribed to SMSI. (ii) It is true that the oxidation state of the metals can change upon illumination, as described in the literature (see, for instance, Piccolo et al., ACS Catalysis 2020, 10, 12696-12705). We have recorded the X-ray photoelectron spectra of the samples after photoreaction to check for changes in the oxidation state of the metals and have found that both Au/TiO₂-001 and Au/TiO₂-101 were reduced under the long-term UV irradiation. The metallic Au species (Au⁰) in Au/TiO₂-001 and Au/TiO₂-101 samples increased from 69.9% and 51.5% to 80.9% and 69.4%, respectively. The oxidation states of Pd components in Pd/TiO₂-001 exhibited similar proportions before and after photoreaction, whereas the Pd/TiO₂-101 also underwent a reduction process.

The change in the oxidation states of metal clusters during the photoreaction can be accounted for the reduction of the metal species induced by the photogenerated electrons under the ethanol/water and UV irradiation condition, which also has been observed in previous studies (Lee et al. Nat. Mater. 18, 620–626 (2019); Piccolo et al. ACS Catal. 2020, 10, 12696–12705). We have added the corresponding discussion in the revised manuscript. In fact, changes in the metal clusters size may facilitate changes in the oxidation states, as the charge transfer between the metal clusters and the titania support is usually related to metallicity. Therefore, changes in metal cluster size and oxidation state can be viewed as simultaneous and both having a potential effect on stability. (iii) We have monitored the temperature of the photocatalysts during the photoreaction and the temperature is always below 54 °C. At this temperature, it seems unlikely the occurrence of sub-surface diffusion, as it has been reported in the seminal study by Schubert et al. (Journal of Materials Chemistry A 2021, 9, 21958). Following the suggestion of the reviewer, we have included this discussion in the revised manuscript (page 7) and the XPS data of the samples after photoreaction in the supplementary information as new Figure S7.

3. The third point is related to several important characterization data:

A. To XPS: XPS analyses were performed for the samples before catalysts,

however – since the paper is focusing on the effect of facets on the stability and activity of the co-catalysts – it would be absolutely necessary to also provide XPS data after catalysis. I am quite sure that photocatalytic conditions (exposure to light) also result in further reduction of Au/Pd/Pt species resulting in a higher proportion of metallic co-catalysts. Importantly, this process (photoreduction) can also be facet-dependent and will also likely contribute to the observed activity trends.

ANSWER: As explained above, we have recorded the XPS data of the samples after the photocatalytic reaction that underwent deactivation (Au and Pd) and included it in the Supplementary Information as new Figure S7. As anticipated by the reviewer, photocatalytic conditions resulted in the partial reduction of the metal clusters. In fact, it is very interesting to see that for both Pt/TiO₂-101 and Pt/TiO₂-001, a progressive improvement of the hydrogen evolution rate is observed (Figure 1). This phenomenon for gas-phase hydrogen evolution from water-ethanol has been described in the literature by Dessal et al. (Journal of Catalysis 2019, 375, 155-163) and later studied by operando X-ray absorption spectroscopy-mass spectrometry by Piccolo et al. (ACS Catalysis 2020, 10, 12696-12705). It has been demonstrated that Pt activation is due to a progressive Pt reduction under photoreaction conditions. We have added a sentence in the revised manuscript regarding the activation of Pt-loaded titania samples under photoreaction (page 5) and included a discussion regarding XPS of Au- and Pd-loaded titania samples (page 7, see above).

B. To XPS discussion (line 136-154): please clarify what you mean by oxidation of Pt/Pd/Au. Do you expect that some of the co-catalyst nanoparticles (e.g. those sitting on a specific facet) are in oxidation state 0 (metallic form), while others are +2/+4 (oxide form). Or do you mean that all and each of the co-catalyst nanoparticles is made of a mixture of oxidation states? In the second case, do you assume that oxidized metals (+2/+4) are at the TiO₂/NP interface or on the NP/air interface? This is important and may need to be considered in both the HER results interpretation and DFT model justification.

ANSWER: It could be certainly great to have an answer to this interesting question posed by the reviewer, but nowadays it is not possible to identify the oxidation state of individual metal clusters on different facets of the same titania crystal. Also, by changing the incident angle of the X-ray source in the XPS experiments (angle-resolved XPS for depth profile analysis), it is not possible to distinguish the oxidation state of the metals at the metal-support interface and at the metal-gas interface, since the mean electron free path is too large to distinguish the different contributions in such small metal clusters.

C. To STEM/HRTEM: all the images presented in the paper are taken from the samples after 1-hour HER runs, however, according to HER profiles in Figure 1,

no (noticeable) deactivation has yet taken place. In fact, the HER deactivation gets prominent only after several hours of light exposure, which raises many questions to the relevance of particle growth to the HER performance.

ANSWER: Thank you very much for this observation. It is a typo; the HAADF-STEM and HRTEM images of the samples after the photoreaction were collected after the stability tests. We are sorry for the inconvenience.

D. To DFT: most of the conclusions drawn in this section are directly related to the fact that Pt has the highest adsorption energy on TiO_2 , which is related to its strongest electronic interaction and results in its highest stability against atomic movement (i.e. sintering) compared to Pd and Au. I do not find the models to be quite appropriate to account for the complexity of the photocatalytic systems, I also do not see the authors considering any experimental data input to shape the calculations/models.

ANSWER: First of all, we would like to clarify that in this study we have performed quantum first-principles simulations (also called *ab initio*) based on density functional theory (DFT). In this type of calculations, the only necessary, and possible, input are the atomic species and the crystal structure of the system. With this information, one goes into solving the Schrödinger equation of the electrons and from its numerical solution it is possible to computationally estimate adsorption energies, charge distributions, band structures, orbitals overlapping, etc. Quantum first-principles simulations are not shaped according to any other additional inputs, in contrast, for instance, to classical molecular dynamics simulations and/or thermodynamic-kinetic models (since we are directly solving the quantum Schrödinger equation of the electronic system, which for being determined it only requires the “names” and positions of the atoms). These features make DFT simulations highly accurate and highly predictive methods, although the computational effort associated to them, obviously, is also quite high. Nevertheless, at present DFT calculations are intensively employed in the field of catalysis with great success since arguably this is one of the few *ab initio* methods that can access the electronic structure of extended materials with affordability and high precision. The high computational cost of our DFT simulations in fact forces us to apply some simplifications on the simulated systems, while keeping the essential ingredients; for instance, we have not considered the likely presence of point defects on the TiO_2 surface, temperature effects have been disregarded, and most of our results have been obtained for single noble metal atoms adsorbed on the TiO_2 surface, instead of clusters. Nevertheless, we would like to emphasize that (1) our DFT simulations quantitatively reproduce most of the experimental observations reported in this study (e.g., the superior stability of Pt atoms on the anatase surface in comparison to those of Au and Pd clusters, both under dark and light conditions), and (2) we have provided plausible arguments showing a correspondence

between the results obtained for noble-metal single atoms and clusters adsorbed on TiO₂ surfaces (see Fig. 5a, new Supplementary Fig.12 and also our response provided to point 7 below).

Moreover, despite that this has not been commented in the manuscript for the sake of focus, our DFT simulations were performed previously to the experiments. In view of our promising theoretical results showing that different TiO₂ facets give rise to different oxidation states in the metal atoms and that the stability of different metal atoms supported on different titania facets were not the same under photoexcitation conditions, the experiments were performed. To our great delight it was found that both types of investigations were fully consistent among them.

Besides these, let me convey a few other less important points, that nevertheless would need clarifications:

4. How sure the authors are that TiO₂ (its phase composition and morphology) did not change upon mechano-activation? At least XRD and BET data should be provided.

ANSWER: We are absolutely sure that the mechanochemical method used for the synthesis did not provoke changes in TiO₂. We have demonstrated this in a previous publication related to the use of mechanochemistry for the preparation of Au/TiO₂ P90 (Applied Materials Today 21 (2020) 100873) and Pd/TiO₂ P90 (Applied Catalysis B: Environmental 309 (2022) 121275), where we did a careful study by XRD, Raman, and BET. We have added this important information and cited this work in the revised manuscript (page 4) and, following the reviewer's suggestion, we have added the XRD, Raman and BET data of the TiO₂-001 and TiO₂-101 before and after the mechanochemical synthesis in the Supplementary Information. Please see Figs. S3, S4 and Table S1. As expected, the anatase TiO₂ crystals were preserved after the ball milling process without any phase transformation of TiO₂ happening during the mechano-activation period.

On the same line, (sorry for picking on the data), after looking at Fig S9: could it be that the TiO₂-001 pattern also features a tiny rutile peak around 27 degrees? I am asking because 500 C treatment could be high enough to facilitate at least a partial A to R transition which needs to be considered.

ANSWER: Based on the reviewer's comment, we have revised the XRD pattern and Raman spectra and have not found evidence for the existence of rutile. The calcination treatment was at 500 °C for only 1 h. See below the enlarged XRD pattern of TiO₂ supports after calcination at 500 °C for 1 h.

5. While the authors make a strong point in the introduction that the choice of the conditions (surface termination, type of co-catalyst, type of reaction) make a strong impact on the photocatalytic performance, their experiments are focused on quite a specific combination of using gas-phase HER and water-EtOH vapor as a reactant. I would suggest that the conclusions derived in this work will likely not apply to a much more typical liquid-media suspension-based photocatalytic HER experiments, and I would thus not agree with the statement in line 89 that they “can be generalized to other types of reactions and probably also to other supports (e.g., rutile TiO₂ and CeO₂).” This, of course, limits the relevance of this research work.

ANSWER: We can not agree neither refute the comment of the reviewer because there is no way at present of knowing whether the results can be extrapolated to photocatalysis in liquid media. We initially included this sentence in the manuscript as an option for future studies; we have now changed the sentence to a more conservative statement (page 3).

6. I am not satisfied with the discussion of the previous research in the introduction. Starting from line 41, the authors jump into the statement that {001} is more efficient than {001}. In a few sentences, they highlight that the choice of the reaction is very important as well as the use of co-catalyst, so why did not they clarify these parameters in their first statement? I generally agree that a lot of factors have to be considered when making such statements, but the introduction makes the impression that nothing has been clarified at all in the literature. In fact, the authors claim in the abstract that “systematic studies on” the “stability during the photoreaction” ... “are absent for metal clusters supported on

TiO₂". This is incorrect and many literature pieces (also cited in the paper) do shed light on the issues.

ANSWER: Following the comment of the reviewer, we have reorganized and improved the introduction section. We are sorry if we have given the impression that we despise the work done previously, it was not our intention! Science is built step by step and based on previous work. For this reason, and having been alerted by the reviewer, we have cautiously revised the introduction to give due credit to previous work.

7. To DFT: how do the DFT models (cluster on TiO₂ and single-atom on TiO₂) correspond to the XPS findings? Do authors think that considering experimental data (oxidation state distribution) is important in order to construct reliable DFT models?

ANSWER: In our DFT calculations, we have found that different types of metal atoms adsorbed on different TiO₂ anatase surfaces exchange different amounts of charge with the substrate, both under dark and photoexcitation conditions (see Figs. 5b,d-i). At the qualitative level, these results are fully consistent with the experimental XPS findings. Now, the main reason why in the XPS experiments a distribution of oxidation states for a given metal species is obtained whereas in the DFT simulations not, is because in the DFT simulations, due to obvious computational limitations, we have simulated single atoms instead of clusters, and within a same metallic cluster the oxidation state of the atoms may strongly vary depending on their relative positions to the anatase surface.

In order to theoretically prove this latter statement, we have performed additional DFT calculations in which we have estimated the charge density difference for an Au⁹ cluster adsorbed on the {001} anatase titania surface with respect to the integrating metallic cluster and TiO₂ surface in isolation (see new Supplementary Fig. 14a). In that figure, it is explicitly shown that different atoms in the same cluster may present quite different oxidation states (i.e., different amount of charge donated or received with respect to an arbitrary reference system), depending on their relative position to the titania substrate. For instance, the two atoms of the cluster located closest to the TiO₂ surface render both the largest amount of donated electronic charge (i.e., the one with largest blue lobules around of it) and received electronic charge (the one with the largest yellow lobules around of it) in the whole cluster. Furthermore, we performed a similar analysis exercise to the calculations presented in Figs.5d-i, in which differences in electronic charge distributions between dark and photoexcitation conditions are represented, but this time considering an Au⁹ cluster adsorbed on the {001} and {101} anatase titania surfaces. Due to the huge computational cost involved in these type of simulations, we could only consider the largest Fermi smearing of 1.70 eV in the photoexcitation simulations (see new Supplementary Fig. 14 b,c).

Therein it is appreciated that the electronic charge distribution differences obtained for an Au⁹ cluster are fully consistent with those found for single Au atoms adsorbed on the same surfaces (Fig. 5f). In particular, under photoexcitation conditions all the atoms in the Au⁹ cluster donate charge to the closest Ti and O substrate atoms as compared to the same system under dark conditions. These new results further corroborate that our DFT conclusions reported in the main text for single metal atom adsorbed on anatase titania surfaces are fully meaningful and representative of the experiments performed for clusters.

Regarding the second Reviewer's question, as we have explained at length in our response to the previous point D, the only possible input in quantum first-principles DFT simulations are the chemical species and crystal structure of the system (i.e., oxidation state distributions cannot be taken as input parameters in this type of advanced quantum calculations, they are actually part of the output in DFT simulations).

8. To DFT, lines 215-217: I do not follow how is this conclusion evident. What is the method to evaluate this? What are the results for Pt? I assume that this result is also obvious from the previous data as Pt was shown to stick to TiO₂ most strongly. This exactly would correspond to the lowest probability of Pt moving from one place to another. Please clarify your method of arriving at this conclusion possibly in the experimental section.

ANSWER: As it appears already explained in the main text and the Methods section, we have employed an effective DFT method that mimic the results of electronic photoexcitation in X-TiO₂. In a nutshell, the employed effective approach consists in constraining the electronic occupation through the Fermi smearing (i.e., to promote electrons from the valence to the conduction band and evade electron-hole recombination) without the need of explicitly simulating photoexcitation. We recall here that a Fermi smearing of σ (eV) in the constrained DFT calculations implies considering the electronic occupation function $f(E,\sigma)=1/(\exp[(E-E_F)/\sigma]+1)$, instead of the usual step function $f(E)=1$ for $E \leq E_F$ and $f(E)=0$ for $E > E_F$, where E_F represents the Fermi energy level. To avoid possible misunderstandings, in the revised version of our manuscript we have added some supplementary explanations about the employed effective DFT approach in the caption of Figure 5 and in the Methods section. Our DFT simulations rendering photoexcitation conditions show that under increasing photoexcitation the preferred adsorption sites for Au and Pd atoms change whereas for Pt remain the same. These results are not obvious since are not a consequence of the fact that Pt sticks most strongly to TiO₂. Metal atoms can be adsorbed on different sites of the anatase TiO₂ surface (for instance, on top of an oxygen atom or in the hollow between oxygen atoms) but one position is always preferred over the others (i.e., the one rendering the largest adsorption energy in absolute value).

All the metallic atoms, Au, Pd and Pt, stick steadily more strongly to the TiO₂ surface under increasing photoexcitation for all the possible adsorption sites (as we have checked, not shown in Fig.5); however, the preferred adsorption site changes under increasing photoexcitation for the Au and Pd atoms but not for Pt atoms. This result is a consequence of the adsorption competition between different possible sites rather than to the adsorption strength on the preferred site. In turn, the competition between different adsorption sites changes under photoexcitation for the Au and Pd metal atoms due to evolving charge redistribution processes occurring on the surfaces, in contrast to the Pt-atom case. In order to avoid possible misinterpretations, in the revised manuscript we have further elaborated on this point in the main text.

Line 56: “it is well known” requires a reference.

ANSWER: Following the comment of the reviewer, we have added the corresponding references.

Line 60: “..many previous studies..” requires a reference or a set of references

ANSWER: Following the comment of the reviewer, we have added the corresponding references.

Line 72: it should rather read “thiolate-protection, and polymer stabilization”

ANSWER: Corrected.

Line 97: I find that the storyline (the Results section starts from HER data) is missing the basic sample characterization details, which I would suggest the authors to highlight briefly at the beginning of the section. Basically, mentioning Fig S8, S9 and the XRF data (not found). I would also strongly advise adding BET values (relevant to my comment 1)

ANSWER: Following the comment of the reviewer, we have included the basic sample characterization details before the HER part in the revised manuscript (Page 4). Also, we have added the BET (Supplementary Table 1) and ICP values (Supplementary Table 2).

Line 98: I would call your “4 (recyclability)” experiment as a “4x1h light on-off cycle experiment”. Recyclability is often associated with the reuse (recovery) of the solid-state catalyst and the notion can thus be misleading.

ANSWER: Corrected.

REVIEWER COMMENTS

Reviewer #1 (Remarks to the Author):

All the concerns are properly addressed in my opinion. It can be published after minor editing.

Reviewer #3 (Remarks to the Author):

I appreciate that the authors did try to address the points raised in my feedback and I am glad they have made the paper more transparent and complete. But, I still find my main concerns (points 1-3) to be valid and not well unaddressed in the manuscript in their essence. These main points still question the relevance/significance of the reported experimental results with regard to the type of conclusions that authors draw from them.

- To be more exact, in your response to point 1, you end by acknowledging my observation, but do not provide an answer to this concern. I still believe that both of your TiO₂ samples have both the facets co-exposed and they shall not be treated (when interpreting the results) as one-facet-exhibiting-only TiO₂ samples. You do mention that the different relative amounts of {001}/{101} interfaces should not contribute to high activity, but should lower it. But this is just a good example of why a sample with higher {001}/{101} interface contribution may behave differently – not only because it shows more/less {001} facet, but because recombination rates are generally much higher. Or, as an example, consider the second half-reaction: MeOH oxidation. Its rate will depend on the abundance of the “second”, HER-inactive facet (assuming facet-dependent e/h separation which is likely). As a result, your HER rates may have nothing to do with the amount of the “HER-active” facet if the entire process is limited by hole extraction and consumption.
- In your response to point 2, you do argue that some of these effects (those I happened to mention, while many others can apply to your system) were not observed (or do not apply) in your investigations. But, the important point is: taking examples of (b) oxidation state change upon illumination or (c) atom diffusion → these effects can be (and likely are) facet-dependent = they will be pronounced to a different extent on your two TiO₂ samples. Because you cannot verify this/quantify their contribution, your assumption that the observed HER trends are a direct representation of the facet-dependent NP action are likely to be shortsighted.
- In your response to point 3, you do not provide any direct response to my main concern (last sentence of my comment).

Let me point out that I do not question the quality and extent of the experimental data that you have provided (they are of outstanding quality), I merely say that the conclusions inferred from the data are not appropriate and do not consider other possibilities and contributions (some of which I suggested in my feedback, but many other effects as well). I am also quite aware that it is probably impossible to prove these points, clarify these effects etc. using the methods currently available to us. Nevertheless, this, unfortunately, makes me question your conclusions.

Reviewer #4 (Remarks to the Author):

The authors reported the systematic study on photocatalytic stability of Pt, Au, and Pd supported on 001 and 101 facets of TiO₂ during hydrogen evolution reaction. The article is well written. However, some points are not clear. I have mentioned that in the comments below. If the authors can address the issues, I would recommend publication in the journal.

Question 1: The author mentioned, "A "Hubbard-U" scheme with $U = 3$ eV was employed for a better treatment of the localized Ti d electronic orbitals (it was checked that the adopted PBE+U setup closely reproduced the experimental lattice parameters of bulk anatase TiO₂, namely, we obtained $a_0 = b_0 = 3.78$ Å and $c_0 = 9.57$ Å)." It is doubtful whether authors used experimentally obtained lattice parameters or DFT-obtained lattice parameters for further calculations. Clarify and explain their rationale behind this.

Question 2: The author should include the electronic band structure profile obtained using PBE-U and HSE06. Otherwise, it is unclear whether $U=3$ eV is able to reproduce the band structure profile.

Question 3: A related concern is whether the authors used Hubbard-U while optimization of noble metal supported on TiO₂. Also, the author should explicitly mention the method adopted for adsorption energy calculation of whether atoms in TiO₂ are allowed to relax.

Question 4: Other point which needs clarification is authors used PBE+U for geometry optimization. However, for electronic properties HSE06 is used. Does this affect the conclusion, if not why.

Question 5. This reviewer's primary concern about the paper is related to the conclusion. It appears that the authors do provide evidence based on DFT calculations about the stability of Pt, Au, and Pd clusters supported on anatase TiO₂ {001} and {101} photocatalysts. However, insight about the higher photocatalytic efficiency of Pt/TiO₂-101 compared to Pt/TiO₂-001 is missing (unless this reviewer missed it). This can be understood by doing calculations related to the actual surface interactions of H₂O, O₂ and necessary intermediated interactions with these surfaces. The suggested revision is for the authors to consider assessing whether such studies are needed as a part of this work and, if not, explain clearly why such studies are not necessary.

Minor:

As a minor point, In figure 3 (k, j) author did not mention the cluster size as done for Figure 3(a). Similar issue is with figure 4 (a,b,c,d).

Suggestion:

Finally, it is observed that the current work would be difficult for other researchers to replicate and confirm, as the submitted manuscript and supporting information do not contain any further calculation details such as, input files, input geometries, etc. In the interest of open science, this reviewer encourages the authors to include such files to make their data findable, accessible, interoperable, and reusable.

Manuscript NCOMMS-22-42901A

ANSWERS TO THE REVIEWERS' COMMENTS

Reviewer #1

All the concerns are properly addressed in my opinion. It can be published after minor editing.

REPLY: We are very grateful for the positive opinion of the Reviewer. We have checked the editing issues.

Reviewer #3

I appreciate that the authors did try to address the points raised in my feedback and I am glad they have made the paper more transparent and complete. But, I still find my main concerns (points 1-3) to be valid and not well unaddressed in the manuscript in their essence. These main points still question the relevance/significance of the reported experimental results with regard to the type of conclusions that authors draw from them.

REPLY: We thank the Reviewer for his/her careful re-evaluation of our work. We would like to stress that we fully agree with his/her observations, and for that reason, we have modified the manuscript following the comments raised. In particular, we have reconsidered some of our statements to avoid any misunderstanding in this second revision as explained below.

However, we would like to recall that, as the Reviewer recognizes at the end of his/her report: "***I am also quite aware that it is probably impossible to prove these points, clarify these effects, etc. using the methods currently available to us.***", his/her concerns are very valuable, but well beyond the possibilities of current science. We would love to have answers to the issues raised by the Reviewer, but it is not currently in our hands. However, we have included appropriate comments about these issues in the re-revised version of our work (see below) to enrich the discussion and, more importantly, to honestly avoid any possible misinterpretation.

- To be more exact, in your response to point 1, you end by acknowledging my observation, but do not provide an answer to this concern. I still believe that both of your TiO₂ samples have both the facets co-exposed and they shall not be treated (when interpreting the results) as one-facet-exhibiting-only TiO₂ samples.

You do mention that the different relative amounts of {001}/{101} interfaces should not contribute to high activity, but should lower it. But this is just a good example of why a sample with higher {001}/{101} interface contribution may behave differently – not only because it shows more/less {001} facet, but because recombination rates are generally much higher. Or, as an example, consider the second half-reaction: MeOH oxidation. Its rate will depend on the abundance of the “second”, HER-inactive facet (assuming facet-dependent e/h separation which is likely). As a result, your HER rates may have nothing to do with the amount of the “HER-active” facet if the entire process is limited by hole extraction and consumption.

REPLY: The Reviewer is absolutely right in the sense that both anatase samples used in our work do contain {001} and {101} facets. In the first revision, we already explained and measured this carefully because it is very important to realize the occurrence of both types of facets. For the sake of clarity, we have introduced a sentence in the re-revised version of the manuscript explaining that the occurrence of both facets may include additional effects that can affect the photocatalytic behavior, such as the {001}/{101} interface contribution (page 5). There are no tools nowadays to precisely demonstrate the exact role of the {001}/{101} interfaces on the photoreaction in the presence/absence of metal clusters. Accordingly, and to avoid any confusion, we have modified the text to substitute the term “facet” with “nanoshape” and “anatase plate” or “anatase bipyramid” when necessary and better explained the occurrence of both facets in the anatase samples prepared (page 4).

- In your response to point 2, you do argue that some of these effects (those I happened to mention, while many others can apply to your system) were not observed (or do not apply) in your investigations. But, the important point is: taking examples of (b) oxidation state change upon illumination or (c) atom diffusion – these effects can be (and likely are) facet-dependent = they will be pronounced to a different extent on your two TiO₂ samples. Because you cannot verify this/quantify their contribution, your assumption that the observed HER trends are a direct representation of the facet-dependent NP action are likely to be shortsighted.

REPLY: Again, we fully agree with the observation of the Reviewer. Particularly stimulating is the discussion about facet-dependent atom diffusion, for instance. But these issues are well beyond the scope of our work and the possibilities of current *operando* characterization techniques. Obviously, we cannot address them in our manuscript. However, we agree with the Reviewer that the assumption that the observed HER trends are a direct representation of the facet-dependent NP action is not straightforward. For that reason, and to address the reader properly, we have included a new sentence in page 5 of the revised

manuscript: “**It is important to recall that the photoactivity trend may be influenced not only by the preferential facet exposed by anatase but also by a more complex situation derived from {001}/{101} interface contributions.**”.

- In your response to point 3, you do not provide any direct response to my main concern (last sentence of my comment).

REPLY: As we already elaborated in our previous response to the previous point “3D. *To DFT: most of the conclusions drawn in this section are directly related to the fact that Pt has the highest adsorption energy on TiO₂, which is related to its strongest electronic interaction and results in its highest stability against atomic movement (i.e., sintering) compared to Pd and Au. I do not find the models to be quite appropriate to account for the complexity of the photocatalytic systems, I also do not see the authors considering any experimental data input to shape the calculations/models.*”, (1) the only necessary and possible input in our first-principles DFT calculations are the atomic species and crystal structure of the simulated systems; in this sense, our DFT simulations do actually take into account the experimental data as input data since we have considered the same TiO₂ polymorph (i.e., anatase), system geometry (i.e., the titania lattice vectors were constrained to their experimental values) and metallic species (i.e., Au, Pd, and Pt) than in the experiments; and (2) in our DFT calculations we have been able to mimic both “dark” and “light-shining” conditions, just like it has been done in the experiments, by using a state-of-the-art *ab initio* approach; in this sense, our work goes beyond what is customarily done in standard computational works, in which only “dark” conditions are simulated. In fact, we have been able to provide very valuable insight into the atomistic mechanisms underlying the experimental findings on the stability of metallic clusters on anatase TiO₂ surfaces.

Furthermore, during the revision of our manuscript, motivated by another Reviewer’s comment, we have also performed additional DFT calculations to understand the causes of the differences in photocatalytic hydrogen activity experimentally observed for the Pt/TiO₂-101 and Pt/TiO₂-001 systems. In particular, we calculated the adsorption free energy of several intermediate molecules involved in the oxidation of water and the reduction of hydrogen molecules (i.e., H*, O*, OH*, and OOH*) on the Pt/TiO₂-101 and Pt/TiO₂-001 surfaces, which allows to theoretically assess the catalytic activity of the analyzed materials for the hydrogen evolution reaction (HER) and oxygen evolution reaction (OER) [see new Supplementary Fig. 15, new discussion in the main text and new computational details in the Methods section]. It was found that in terms of HER performance the two Pt/TiO₂-101 and Pt/TiO₂-001 systems are pretty similar, and in principle very efficient, since the estimated proton adsorption free

energies, ΔG_{H^*} , are quite small in absolute value in both cases (i.e., 0.15-0.20 eV) and very similar in size to that of the archetypal HER catalyst Pt (111). On the other hand, in terms of OER activity, the Pt/TiO₂-101 system turns out to be much superior than the Pt/TiO₂-001 system since the OER overpotential estimated for the former system is much smaller and comparable in size to those of archetypal OER catalysts like SrCoO₃, RuO₂, and PtO₂. Therefore, based on these new DFT calculations, we can actually rationalize the likely causes for the observed higher photocatalytic efficiency in H₂ production of Pt/TiO₂-101 in comparison to that of Pt/TiO₂-001: in terms of the oxygen evolution reaction (OER), the former system is a much more efficient catalyst.

- Let me point out that I do not question the quality and extent of the experimental data that you have provided (they are of outstanding quality), I merely say that the conclusions inferred from the data are not appropriate and do not consider other possibilities and contributions (some of which I suggested in my feedback, but many other effects as well). I am also quite aware that it is probably impossible to prove these points, clarify these effects etc. using the methods currently available to us. Nevertheless, this, unfortunately, makes me question your conclusions.

REPLY: As discussed above, we agree with the Reviewer that current state-of-the-art characterization capabilities are not able to answer his/her stimulating questions. Following the comments of the Reviewer, we have modified the manuscript's text as explained above to constrain our work's conclusions. We hope that the Reviewer will appreciate our effort.

Reviewer #4

The authors reported the systematic study on photocatalytic stability of Pt, Au, and Pd supported on 001 and 101 facets of TiO₂ during hydrogen evolution reaction. The article is well written. However, some points are not clear. I have mentioned that in the comments below. If the authors can address the issues, I would recommend publication in the journal.

REPLY: We thank the Reviewer for his/her careful evaluation of our work and very useful and insightful comments.

Question 1: The author mentioned, "A "Hubbard-U" scheme with U = 3 eV was employed for a better treatment of the localized Ti d electronic orbitals (it was checked that the adopted PBE+U setup closely reproduced the experimental

lattice parameters of bulk anatase TiO₂, namely, we obtained $a_0 = b_0 = 3.78 \text{ \AA}$ and $c_0 = 9.57 \text{ \AA}$.” It is doubtful whether authors used experimentally obtained lattice parameters or DFT-obtained lattice parameters for further calculations. Clarify and explain their rationale behind this.

REPLY: We thank the Reviewer for drawing our attention to these important technical aspects of our DFT calculations. Thanks to his/her comment we have realized that there were some unfortunate errors in our previous Methods description.

In our DFT calculations, the lattice vectors of anatase were fixed to the corresponding experimental values, namely, $a_0=b_0=3.78 \text{ \AA}$ and $c_0=9.51 \text{ \AA}$ (J. Am. Chem. Soc. 109, 3639 -1987-). The reason for this choice is that the PBE+U approach cannot simultaneously reproduce the experimental lattice structure and band gap of anatase TiO₂ (i.e., $E_g=3.2 \text{ eV}$).

Such a PBE+U limitation is clearly appreciated in Figures (a) and (b) above: in order to obtain a theoretical bandgap close to the experimental value of 3.2 eV one needs to increase the size of the U parameter beyond a reasonable value of 9 eV (Figure b); however, upon increasing the value of U the obtained relaxed lattice parameters increasingly depart from the experimental a_0 and c_0 values (Figure a). Similar test calculations were performed for the PBEsol+U exchange-correlation functional (not shown here), arriving at the same conclusions.

All the DFT geometry relaxations involved in the present work were performed at the PBE+U level with $U=3$ eV and by constraining the size of the lattice vectors to their experimental values; the positions of the atoms, however, were allowed to fully relax in all the cases. Subsequently, the energy, charge density, and optoelectronic properties of anatase TiO_2 were estimated with the range-separated hybrid HSE06 exchange-correlation functional without performing further relaxations.

The reason for adopting this computational approach is twofold. First, full geometry relaxations of the slabs simulated in this study are not computationally feasible with the hybrid HSE06 exchange-correlation functional. And second, from Figure c above it is clearly appreciated that practically any anatase geometry generated at the PBE+U level, by constraining the lattice vectors to their experimental values, renders a HSE06 band gap that is in excellent agreement with the experimental value of 3.2 eV.

Therefore, it can be concluded that the adopted computational approach is very robust because (1) it reproduces the structural and optoelectronic properties of anatase TiO_2 as they have been measured in experiments, and (2) the specific value of U that is adopted for the calculations has a very minor impact on the obtained optoelectronic properties.

In the revised version of our manuscript, we have corrected all the errors that were present in our original first-principles Methods description and added to the Supplementary Information a new discussion on the rationale behind the selection of the employed DFT approach (e.g., see the new “First-principles Simulation Techniques” section in the Supplementary Information and the new Supplementary Figure 18).

Question 2: The author should include the electronic band structure profile obtained using PBE-U and HSE06. Otherwise, it is unclear whether $U=3$ eV is able to reproduce the band structure profile.

REPLY: Following the Reviewer’s suggestion, in the revised version of our manuscript now we present the total and partial density of states (pDOS) obtained for anatase TiO_2 as calculated with the PBE+U ($U=3$ eV) and range-separated hybrid HSE06 exchange-correlation functionals (Figure d in our response to Question 1 above). In Figure d above, it is appreciated that the band gap obtained at the PBE+U level is smaller than that obtained with the more accurate hybrid HSE06 functional because the former functional tends to underestimate the energy difference between the top of the conduction band and the Fermi energy level. On the other hand, the partial pDOS contributions stemming from the O p and Ti d electronic orbitals to both the top of the valence band and the bottom of

the conduction band are very similar as obtained with the two different methods. In the revised version of our manuscript, this new information appears explained in the extended “First-principles Simulation Techniques” section in the Supplementary Information and the new Supplementary Figure 18.

Nevertheless, as we have mentioned in the previous point and now appears clarified in the main text, all the energy, charge density, and optoelectronic DFT results presented in this study were estimated with the range-separated hybrid HSE06 exchange-correlation functional by considering the geometries generated at the PBE+U level. Therefore, in spite of the fact that the PBE+U (U=3 eV) approach is not able to fully reproduce the electronic band structure obtained with the hybrid HSE06 exchange-correlation functional, that limitation does not affect the main theoretical conclusions presented in our work.

Question 3: A related concern is whether the authors used Hubbard-U while optimization of noble metal supported on TiO₂. Also, the author should explicitly mention the method adopted for adsorption energy calculation of whether atoms in TiO₂ are allowed to relax.

REPLY: Regarding our DFT calculations for the noble metal ions supported on anatase TiO₂, we should clarify that: (1) the PBE+U (U=3 eV) approach was employed to perform all the geometry relaxations and the U energy parameter exclusively acts on the Ti *d* orbitals, (2) in the geometry relaxations, the dimensions of the simulated supercell were kept fixed (also due to the presence of a vacuum region) while the positions of all the atoms were allowed to vary so as to render vanishingly small atomic forces, (3) the adsorption energies were estimated with the range-separated hybrid HSE06 exchange-correlation functional by considering the geometries generated at the PBE+U level. In the revised version of our manuscript, this information now appears explicitly explained in the Methods and Supplementary “First-principles Simulation Techniques” sections.

Question 4: Other point which needs clarification is authors used PBE+U for geometry optimization. However, for electronic properties HSE06 is used. Does this affect the conclusion, if not why.

REPLY: To answer this Reviewer’s point, we recall our response given to Question 1 above. In particular, we adopted this computational approach because, besides being computationally feasible, (1) it reproduces both the structural and optoelectronic properties of anatase TiO₂ as they have been measured in experiments, and (2) the specific value of U that is adopted in the calculations has a very minor impact on the obtained optoelectronic properties.

This rationale is explained in the revised “First-principles Simulation Techniques” section in the Supplementary Information and in the new Supplementary Figure 18.

Question 5. This reviewer's primary concern about the paper is related to the conclusion. It appears that the authors do provide evidence based on DFT calculations about the stability of Pt, Au, and Pd clusters supported on anatase TiO₂ {001} and {101} photocatalysts. However, insight about the higher photocatalytic efficiency of Pt/TiO₂-101 compared to Pt/TiO₂-001 is missing (unless this reviewer missed it). This can be understood by doing calculations related to the actual surface interactions of H₂O, O₂ and necessary intermediated interactions with these surfaces. The suggested revision is for the authors to consider assessing whether such studies are needed as a part of this work and, if not, explain clearly why such studies are not necessary.

REPLY: The original aim of performing the DFT first-principles calculations presented in this study was to obtain insights into the stability of metallic clusters on different anatase TiO₂ surfaces under dark and light-shining conditions, not into the photocatalytic efficiency differences of different anatase TiO₂ surfaces. Nevertheless, in view of the Reviewer's comment, we have performed additional DFT calculations in order to attempt to understand the differences in photocatalytic hydrogen activity between the Pt/TiO₂-101 and Pt/TiO₂-001 systems.

For these new DFT calculations, we essentially followed the computational strategies explained in the previous works [J. Mater. Chem. A 10, 18132 (2022)], [ACS Sustainable Chem. Eng. 10, 16924 (2022)], [J. Phys. Chem. C 118, 4095 (2014)] and [ChemCatChem 3, 1159 (2011)], since they allow to effectively estimate the performance of catalytic materials for the hydrogen evolution (HER) and oxygen evolution (OER) reactions. In particular, we calculated the adsorption free energy of several intermediate molecules involved in the oxidation of water and the reduction of hydrogen molecules (i.e., H*, O*, OH*, and OOH*) on the Pt/TiO₂-101 and Pt/TiO₂-001 surfaces. The main results of our new H₂ production catalytic DFT calculations are shown in the Figure below.

As it can be appreciated in Figure a, in terms of HER performance the two Pt/TiO₂-101 and Pt/TiO₂-001 systems are pretty similar, and in principle very efficient, since the estimated proton adsorption free energies, ΔG_{H^*} , are quite small in absolute value in both cases (i.e., 0.15-0.20 eV) and very similar in size to that of the archetypal HER catalyst Pt (111). On the other hand, in terms of OER activity, the Pt/TiO₂-101 system turns out to be much superior than the Pt/TiO₂-001 system since the OER overpotential estimated for the former system is much smaller and comparable in size to those of archetypal OER catalysts like SrCoO₃, RuO₂, and PtO₂ (Figure b). Therefore, based on these new DFT calculations, we can actually rationalize the likely causes for the observed higher photocatalytic efficiency in H₂ production of Pt/TiO₂-101 in comparison to that of Pt/TiO₂-001: in terms of the oxygen evolution reaction (OER), the former system is a much more efficient catalyst.

In the revised version of our article, the results of these new DFT calculations and the corresponding technical parameters appear explained in the main text (before the Conclusions and in the Methods sections) and in the Supplementary Information (new Supplementary Fig. 15). Likewise, we have added some relevant bibliographic references to the main text.

Minor:

As a minor point, in figure 3 (k, j) author did not mention the cluster size as done for Figure 3(a). Similar issue is with figure 4 (a,b,c,d).

REPLY: In Figure 3 (k, j), the sub-nanometer Pt clusters were too small and escaped detection with HAADF-STEM. For this reason, we used HRTEM in addition to HAADF-STEM to confirm the existence and size of the Pt clusters for both Pt/TiO₂-001 and Pt/TiO₂-101 (see Figs. S8 and S9). In Figure 4 (a, b, c, and d), the Au particles were quite large and it was difficult to collect enough particles in HRTEM to count the size distribution (at least one hundred particles), so we

used HAADF-STEM to determine the size with good statistics. Therefore, due to these uncertainties, we only decided to include the cluster size in Figures 3 and 4 when appropriate.

Suggestion:

Finally, it is observed that the current work would be difficult for other researchers to replicate and confirm, as the submitted manuscript and supporting information do not contain any further calculation details such as, input files, input geometries, etc. In the interest of open science, this reviewer encourages the authors to include such files to make their data findable, accessible, interoperable, and reusable.

REPLY: Following the recommendation of the Reviewer, we have compiled the input and output files employed and produced, respectively, in the first-principles density functional theory (DFT) calculations presented in this study in a public repository at:

<https://dataverse.csuc.cat/dataset.xhtml?persistentId=doi:10.34810/data756>.

By doing this, we are making our data accessible, interoperable, and reusable to any interested reader in the interest of Open Science. Such input and output files follow the format of the DFT software VASP (<https://www.vasp.at/>). It is noted that some of the necessary input files for carrying out the VASP calculations (e.g., POTCAR) have not been included in our dataset in order to avoid possible legal issues (i.e., the use of VASP requires a license agreement). We have included this information in the Data availability statement in the revised manuscript.

REVIEWER COMMENTS

Reviewer #1 (Remarks to the Author):

Some concerns are not fully addressed yet, but it is partly a limitation of current analytical techniques as the reviewer noted. This revised manuscript has further been improved. I do not find any critical issues that can discourage this work from being published.

Reviewer #4 (Remarks to the Author):

The authors have answered all the concerns of the reviewers. I recommend it now for publication.

Reviewer #5 (Remarks to the Author):

As a newly invited reviewer for the revised MS, I do not believe the MS is suitable for a prestigious journal such as Nature Communications. Traditional noble metal nanoparticles are losing relevance as co-catalysts due to massive achievements in single atom photocatalysis field. The presented results on the nanoparticles stability on the common (001) or (101) TiO₂ facets have already been investigated and the findings are not of a high significance, and are certainly not sufficient for Nature Communications.

The authors have responded to the comments of the Reviewer #3, mostly by acknowledging the uncertainties in the experiments, but fundamental issues still remain:

1) I have some doubts that all the noble metal precursors can fully react with TiO₂ during the dry ball-milling for 10 min. The unreacted precursors could lead to photodeposition of noble metals during photocatalytic tests – this would likely discredit the presented stability tests. Non-metallic XPS peaks observed in Figure 2 could be partially due to the remaining precursor species.

2) Better quality HAADF-STEM images (and STEM-EDX elemental mapping) should be taken to confirm, if atomic clusters or single atoms are present on the surface.

3) XPS for Pt-TiO₂ after photocatalytic tests should be provided.

4) The provided TEM and HAADF-STEM do not allow to make clear conclusions about stability of NPs. Overall better sample preparation and characterization are required for such studies to be convincing. Specific examples:

a. In Figure S8-9 it is hard confirm the existence of the particles indicated by the arrows. These features look like surface roughness.

b. Figure 3 is hard to analyze due to low magnification/resolution, and overlap between particles. For the facet-specific studies, a much more clear characterization with distinguishable facets is required.

c. Figure 3g seemingly shows larger particles, than Figure 3h.

d. Figure 3i-l – I cannot see any sign of Pt particles (which should be apparent by HAADF-STEM).

5) I have some doubt about the ability of dry ball-milling method to produce sub-nanometer surface metal clusters with such a narrow size distribution. I'm not convinced that the provided images before and after reaction are representative – the authors should demonstrate this in multiple images.

6) The photoreactor design is quite unusual – I wonder if it allows for reproducible results. It seems very uncertain if a reproducible particle distribution on the cellulose paper could be achieved in every test, or if a similar temperature could be maintained for all photocatalysts.

7) The stability of cellulose structure during drying/wetting, UV-exposure and gas permeation is questionable. Structural changes of the photocatalyst supported on cellulose could lead to the decay of H₂ evolution.

8) "These ultra-small metal entities are prone to aggregate when irradiated with light due to their high surface free energy, thereby reducing their surface-to-bulk atoms ratio and, in turn, photocatalytic performance. [30]" – This statement can be misleading, as typically, not only light but reductive conditions are required for aggregation [Current Opinion in Chemical Engineering 2023, 40:100921].

The study [30] seems to be unrelated to this statement and should not be cited here.

Reviewer #5

1) I have some doubts that all the noble metal precursors can fully react with TiO₂ during the dry ball-milling for 10 min. The unreacted precursors could lead to photodeposition of noble metals during photocatalytic tests – this would likely discredit the presented stability tests. Non-metallic XPS peaks observed in Figure 2 could be partially due to the remaining precursor species.

REPLY: We can ensure that all the noble metal precursors react with the anatase support during ball milling because no signals of the precursors (acetate or acetylacetonate) are detected by either Raman spectroscopy or XPS (C 1s signal) in the samples prepared. For the sake of clarity, we have added a sentence in this regard in the “Methods” section (pages 14-15).

2) Better quality HAADF-STEM images (and STEM-EDX elemental mapping) should be taken to confirm, if atomic clusters or single atoms are present on the surface.

REPLY: We have already discussed this issue in the previous answers to Reviewers #3 and #4. Nevertheless, and following the recommendation of the Reviewer, we have included additional, representative HAADF-STEM-EDX for Pd/TiO₂ as new Supplementary Figure 12 to clearly show the correlation between the metal and the structures identified as metal clusters in the samples. Accordingly, we have added a sentence in Page 8 of the revised manuscript. As discussed with previous referees, we do not have better instrumentation to quantitatively discuss the occurrence of exclusively atomic clusters vs. single atoms.

Supplementary Figure 12. Fresh (a) Pd/TiO₂-001 and (b) Pd/TiO₂-101

3) XPS for Pt-TiO₂ after photocatalytic tests should be provided.

REPLY: Following the comment of the reviewer, we have included the XPS for Pt-TiO₂ after photoreaction in the Supplementary Figure 7 and added the corresponding sentence in the revised manuscript (page 7).

Supplementary Figure 7. XPS results of Au/TiO₂, Pd/TiO₂ and Pt/TiO₂ photocatalysts after 20 hours of photoreaction. XP Au 4f (a), Pd 3d (b) and Pt 4f (c) spectra of post-reacted Au/TiO₂-001, Au/TiO₂-101, Pd/TiO₂-001, Pd/TiO₂-101, Pt/TiO₂-001 and Pt/TiO₂-101 photocatalysts. The corresponding ratios of metal oxidation states extracted from XP spectra in (a), (b) and (c) are shown in (d), (e) and (f), respectively.

4) The provided TEM and HAADF-STEM do not allow to make clear conclusions about stability of NPs. Overall better sample preparation and characterization are required for such studies to be convincing. Specific examples:

a. In Figure S8-9 it is hard confirm the existence of the particles indicated by the arrows. These features look like surface roughness.

b. Figure 3 is hard to analyze due to low magnification/resolution, and overlap between particles. For the facet-specific studies, a much more clear characterization with distinguishable facets is required.

c. Figure 3g seemingly shows larger particles, than Figure 3h.

d. Figure 3i-l – I cannot see any sign of Pt particles (which should be apparent by HAADF-STEM).

REPLY: We are sorry for the inconvenience. For a clear visualization, and according to the comment of the Reviewer, we have added new Supplementary Figures 8 and 9 with additional HAADF-STEM and HRTEM images of the samples and cited them accordingly in the revised text. Regarding Figures 3g, 3h and 3i-l, we would like to recall that we used HRTEM in addition to HAADF-STEM (Fig. 4, new Figs. S8 and S9, and Figs. S10 and S11,) to better determine the size of the subnanometric entities, as already explained in the original text (page 8).

Supplementary Figure 8. HAADF-STEM images of fresh (a, c, e, g, i, k) and after reaction (b, d, f, h, j, l) samples. a,b, Au/TiO₂-001. c,d, Au/TiO₂-101. e,f, Pd/TiO₂-001. g,h, Pd/TiO₂-101. i,j, Pt/TiO₂-001. k,l, Pt/TiO₂-101. Scale bar: 10 nm. The bright dots refer to the corresponding metal clusters in each photocatalyst.

Supplementary Figure 9. HRTEM images of the metal clusters of the fresh and after photoreaction samples. a,b, Au/TiO₂-001. c,d, Au/TiO₂-101. e,f, Pd/TiO₂-001. g,h, Pd/TiO₂-101. i,j, Pt/TiO₂-001. k,l, Pt/TiO₂-101. Scale bar: 5 nm. Metal clusters are indicated by arrows.

5) I have some doubt about the ability of dry ball-milling method to produce sub-nanometer surface metal clusters with such a narrow size distribution. I'm not convinced that the provided images before and after reaction are representative – the authors should demonstrate this in multiple images.

REPLY: The formation of subnanometer metal clusters in a narrow size distribution by mechanochemistry is well established and has been widely reported in the literature (see, for instance, He, X. et al. *Cell Rep. Phys. Sci.* 1, 100004 (2020); Hu, Y. et al. *Green Chem.* 23, 8754 (2021); Hu, Y. et al. *Mat. Today* 63, 288 (2023); Chen, Y. et al. *Appl. Catal. B Environ.* 309, 121275 (2022); Chen, Y. et al. *Appl. Mater. Today* 21, 100873 (2020)). In these works, the presence of subnanometer metal clusters has been demonstrated not only by transmission electron microscopy techniques, but also by using photoemission spectroscopy and X-ray absorption. Our images recorded over the fresh and the used samples are fully representative, and the size distribution histograms have been obtained by measuring hundreds of particles, as it is usually done in heterogeneous catalysis studies. As explained above, we have included additional HAADF-STEM and HRTEM images as new Supplementary Figures 8 and 9.

6) The photoreactor design is quite unusual – I wonder if it allows for reproducible results. It seems very uncertain if a reproducible particle distribution on the cellulose paper could be achieved in every test, or if a similar temperature could be maintained for all photocatalysts.

REPLY: Our photoreactor design has, in fact, many advantages. It allows to run the photoreaction in gas phase and under continuous flow operation, which is much better than the conventional slurry reactors in terms of mass transfer limitations and photon delivery. We have discussed this in many of our previous works (see, for instance, Torras et al. *Adv. Energy Materials* 12, 2103733 (2022)). The reproducibility of our results is outstanding as we have repeated systematically many of our experiments with hydrogen photoreaction rates virtually identical. Fig. 1 and Fig S5 show that, for the same M/TiO₂ sample, three different experiments were carried out, and they exhibited exactly the same photocatalytic behaviour in terms of activity and stability. Regarding the temperature, we measure the temperature directly on the sample with a thermocouple. We have checked the position of the thermocouple on different parts of the sample and the temperature value is virtually identical because the sample is very well dispersed on the cellulose paper, which represents another important advantage of our photoreactor design. Finally, we monitored the temperature of the photoreactor during different long-term photocatalytic reactions, and it was always 50–52 °C (see below for Au/TiO₂, Pd/TiO₂ and Pt/TiO₂). Therefore, we can safely conclude that our experimental results are fully reproducible.

7) The stability of cellulose structure during drying/wetting, UV-exposure and gas permeation is questionable. Structural changes of the photocatalyst supported on cellulose could lead to the decay of H₂ evolution.

REPLY: Previous to this work, we have already carefully checked the stability of the cellulose paper after 100 h of photoreaction, both in blank experiments and after impregnation with the photocatalysts, by FTIR spectroscopy, Raman spectroscopy, X-ray photoelectron spectroscopy and gas permeation experiments. In all cases, the results demonstrate that the cellulose paper is perfectly stable under the operation conditions of the photocatalytic experiments and there are no structural changes. For the sake of clarity, we have included a sentence in the “Methods” section (page 16).

8) “These ultra-small metal entities are prone to aggregate when irradiated with light due to their high surface free energy, thereby reducing their surface-to-bulk atoms ratio and, in turn, photocatalytic performance. [30]” – This statement can be misleading, as typically, not only light but reductive conditions are required for aggregation [Current Opinion in Chemical Engineering 2023, 40:100921]. The study [30] seems to be unrelated to this statement and should not be cited here.

REPLY: We are grateful to the Reviewer for this observation. Accordingly, we have modified the statement as “These ultra-small metal entities are prone to aggregate when irradiated with light under reductive conditions due to their high surface free energy, thereby reducing their surface-to-bulk atoms ratio and, in turn, photocatalytic performance. [30]” in page 3 of the revised manuscript and substituted reference [30] with the article “Deciphering the issue of single-atom catalyst stability” by Kali Rigby and Jae-Hong Kim in Current Opinion in Chemical Engineering 2023, 40:100921.